# Comparative genomic analysis reveals metabolic flexibility of Woesearchaeota

Wen-Cong Huang[1,7], Yang Liu [1,7], Xinxu Zhang[1], Cui-Jing Zhang[1], Dayu Zou[1,2], Shiling Zheng[3], Wei Xu [4], Zhuhua Luo [4,5], Fanghua Liu [3,6] & Meng Li [1✉]

The archaeal phylum Woesearchaeota, within the DPANN superphylum, includes phylogenetically diverse microorganisms that inhabit various environments. Their biology is poorly understood due to the lack of cultured isolates. Here, we analyze datasets of Woesearchaeota 16S rRNA gene sequences and metagenome-assembled genomes to infer global distribution patterns, ecological preferences and metabolic capabilities. Phylogenomic analyses indicate that the phylum can be classified into ten subgroups, termed A–J. While a symbiotic lifestyle is predicted for most, some members of subgroup J might be host-independent. The genomes of several Woesearchaeota, including subgroup J, encode putative [FeFe] hydrogenases (known to be important for fermentation in other organisms), suggesting that these archaea might be anaerobic fermentative heterotrophs.

[1] Shenzhen Key Laboratory of Marine Microbiome Engineering, Institute for Advanced Study, Shenzhen University, Shenzhen, China. [2] Department of Ocean Science, The Hong Kong University of Science and Technology, Hong Kong, China. [3] Key Laboratory of Coastal Biology and Biological Resources Utilization, CAS Key Laboratory of Coastal Environmental Processes and Ecological Remediation, Yantai Institute of Coastal Zone Research, Chinese Academy of Sciences, Yantai, China. [4] Key Laboratory of Marine Biogenetic Resources, Third Institute of Oceanography, Ministry of Natural Resources, Xiamen, China. [5] School of Marine Sciences, Nanjing University of Information Science & Technology, Nanjing, China. [6] National-Regional Joint Engineering Research Center for Soil Pollution Control and Remediation in South China, Guangdong Key Laboratory of Integrated Agro-environmental Pollution Control and Management, Institute of Eco-environmental and Soil Sciences, Guangdong Academy of Sciences, Guangzhou, China. [7] These authors contributed equally: Wen-Cong Huang, Yang Liu. ✉email: limeng848@szu.edu.cn

The known scope of archaeal diversity has noticeably expanded in recent years after the discovery of novel lineages, enabled by the development of bioinformatics methodologies, the continually generated sequencing data, and cultivation. The expansion of the archaeal tree of life has changed the picture of the ecological and evolutionary importance of archaea. For example, we now know that Thaumarchaeota, ammonia oxidizers detected in aquatic and terrestrial environments, participate in the global nitrogen cycle[1]; that some members of Bathyarchaeota, archaea from a non-euryarchaeal lineage, exhibit methanogenic characteristics[2,3]; that the genomic content of Asgard archaea sheds light on the origin of eukaryotes[4,5]; and that the DPANN (Diapherotrites, Parvarchaeota, Aenigmarchaeota, Nanoarchaeota, Nanohaloarchaeota) archaea, a proposed monophyletic group of enigmatic archaea, are typically small cells harboring reduced genomes with a limited metabolic repertoire[6,7]. Notably, few DPANN members have been successfully enriched in co-culture and reportedly rely upon their hosts to proliferate[8–10]. For instance, *Nanoarchaeum equitans* is an obligate ectosymbiont of the host, *Ignicoccus hospitalis*, which provides growth factors, lipids, amino acids, and probably ATP, to *N. equitans*[8]. However, in a recent study, researchers generated single amplified genomes affiliated with DPANN lineage using fluorescence-activated cell sorting. They found minor heterogeneous DNA sources from potential hosts in the genomes, which raised the possibility that most DPANN archaea might not lead a symbiotic lifestyle in subsurface environments[11].

The Woesearchaeota phylum (formerly Euryarchaea DHVEG-6) was proposed within the DPANN superphylum in 2015[6]. Woesearchaeota are ubiquitous residents of various environments (e.g., groundwater[6], soil[12], marine sediments[13], hydrothermal vents[14], and freshwater sediments[15]) where they may shape the surroundings and impact global biogeochemical cycles, interacting or not with other organisms. For instance, based on genome-resolved metagenomic analysis, Castelle et al.[6] proposed that Woesearchaeota AR20 may lead a symbiotic lifestyle, and are involved in anaerobic carbon and hydrogen cycles. Later, Castelle and Banfield[16] reported that some Woesearchaeota may employ the bacterial methylerythritol phosphate (MEP) pathway transferred from Firmicutes to synthesize isopentenyl pyrophosphate and dimethylallyl diphosphate precursors for cell membrane assembly. Investigation of approximately 1000 genomes reconstructed from several metagenomics-based studies revealed that a Woesearchaeota genome, with features suggesting a fermentation-based lifestyle, encodes a near-complete glycolysis pathway, components of a potential metal-reducing respiratory pathway involved in iron metabolism, and a cytoplasmic MvhD-HdrABC complex (F420-non-reducing hydrogenase iron-sulfur subunit D and heterodisulfide reductase ABC) functioning in the final step of methanogenic pathways[17]. Collectively, these studies suggest intriguing metabolic diversity among the Woesearchaeota.

Liu et al.[18] reported a co-occurrence pattern between the operational taxonomic units (OTUs) affiliated with Woesearchaeota 16S rRNA genes, and those of Methanomicrobia and Methanobacteria, which indicated possible interactions between members of these groups. Based on these findings, the authors proposed that Woesearchaeota probably provide substrates for $H_2/CO_2$-utilizing methanogens and acetate-utilizing methanogens in return for amino acids and other compounds, to compensate for their own metabolic deficiencies. Meanwhile, a positive correlation of the Woesearchaeota relative abundance and bacterial community was reported in a study based on 16S rRNA gene amplicon sequences, suggesting possible interactions between these microbes[19].

Despite the above-mentioned glimpses into the metabolic potential of Woesearchaeota archaea and interactions with other organisms, their ecological patterns, metabolic diversity, and evolutionary history remain unclear. To address that, here, we retrieved the Woesearchaeota 16S rRNA gene sequences from the Earth Microbiome Project (EMP) datasets. We then analyzed genomes of Woesearchaeota from different environments, including 152 metagenome-assembled genomes (MAGs), with 49 MAGs reported for the first time in the current study. Our analyses help us to understand the global distribution patterns of Woesearchaeota in different biotopes, and shed new light on the metabolism and evolutionary history of Woesearchaeota diversification.

## Results

**Ecological patterns of Woesearchaeota distribution.** To survey the global distribution and abundance of Woesearchaeota, we selected 2,163 16S rRNA gene libraries from EMP amplicon datasets, using the criteria: minimum sequencing depth of 30,000 and minimum relative abundance of Woesearchaeota in the libraries of 0.1%. These libraries were sampled from 11 distinct biotope types worldwide, with the relative Woesearchaeota abundance ranging from 0.1 to 3.9% (Fig. 1a). The relative abundance of Woesearchaeota in saltmarshes was significantly higher than that in other biotopes ($p < 0.01$, *post hoc* test after ANOVA; Supplementary Fig. 1). The alpha diversity (PD_whole_tree value) of Woesearchaeota was significantly higher in saltmarshes, freshwater, and mangrove than in other biotopes, with the lowest value in the sand samples (Fig. 1b). The t-distributed stochastic neighbor embedding (t-SNE) analysis of the similarity of Woesearchaeota community matrix (the unweighted UniFrac metric) revealed that the saline biotopes were separated from plant and non-saline biotopes (Fig. 1c; $F = 173.78$, $p = 0.001$, $R^2 = 0.138$, 999-permutations PERMANOVA test). Further, communities in different biotopes were significantly different from each other (Fig. 1d; $F = 103.72$, $p = 0.001$, $R^2 = 0.325$). Finally, by testing correlations between the available physicochemical parameters and beta diversity of Woesearchaeota communities using the Mantel test, we discovered that salinity levels were significantly correlated with Woesearchaeota community composition ($r = 0.295$, $p = 0.001$, $n = 167$) (Supplementary Table 1). These observations indicate that salinity influences Woesearchaeota community and distribution.

**Woesearchaeota genome dataset.** We reconstructed 49 metagenome-assembled genomes (MAGs) affiliated with Woesearchaeota. These MAGs were recovered from a wide range of environments, including various water depths of the Yap trench, intertidal mangrove sediments of Mai Po Nature Reserve (Hong Kong), Futian Mangrove Nature Reserve (Shenzhen), seagrass sediments of Swan Lake Nature Reserve (Rongcheng) and sediments of Jiulong River estuary (Fujian). The MAGs reconstructed here were estimated to be 77.7% (median) complete and 1.5% (median) contaminated (Supplementary Fig. 2). These MAGs were combined with 103 publicly available genomes (estimated completeness ≥50% and contamination ≤10%), resulting in a dataset of 152 MAGs (Supplementary Data 1).

**Distinct Woesearchaeota subgroups and their metabolic potential.** Maximum-likelihood phylogenetic tree based on the carefully selected single-copy orthologs (see "Methods"), with most nodes with ultrafast bootstrap (UFBOOT) values ≥95% and Shimodaira and Hasegawa-like approximate likelihood ratio test (SH-aLRT) values ≥80%, divided Woesearchaeota into ten phylogenetically distinct subgroups (subgroups A–J; Fig. 2). Most subgroups are monophyletic in the phylogenetic trees based on the 16S rRNA gene and 15 ribosomal proteins (see Supplementary Notes). In six subgroups (C–J), the average estimated

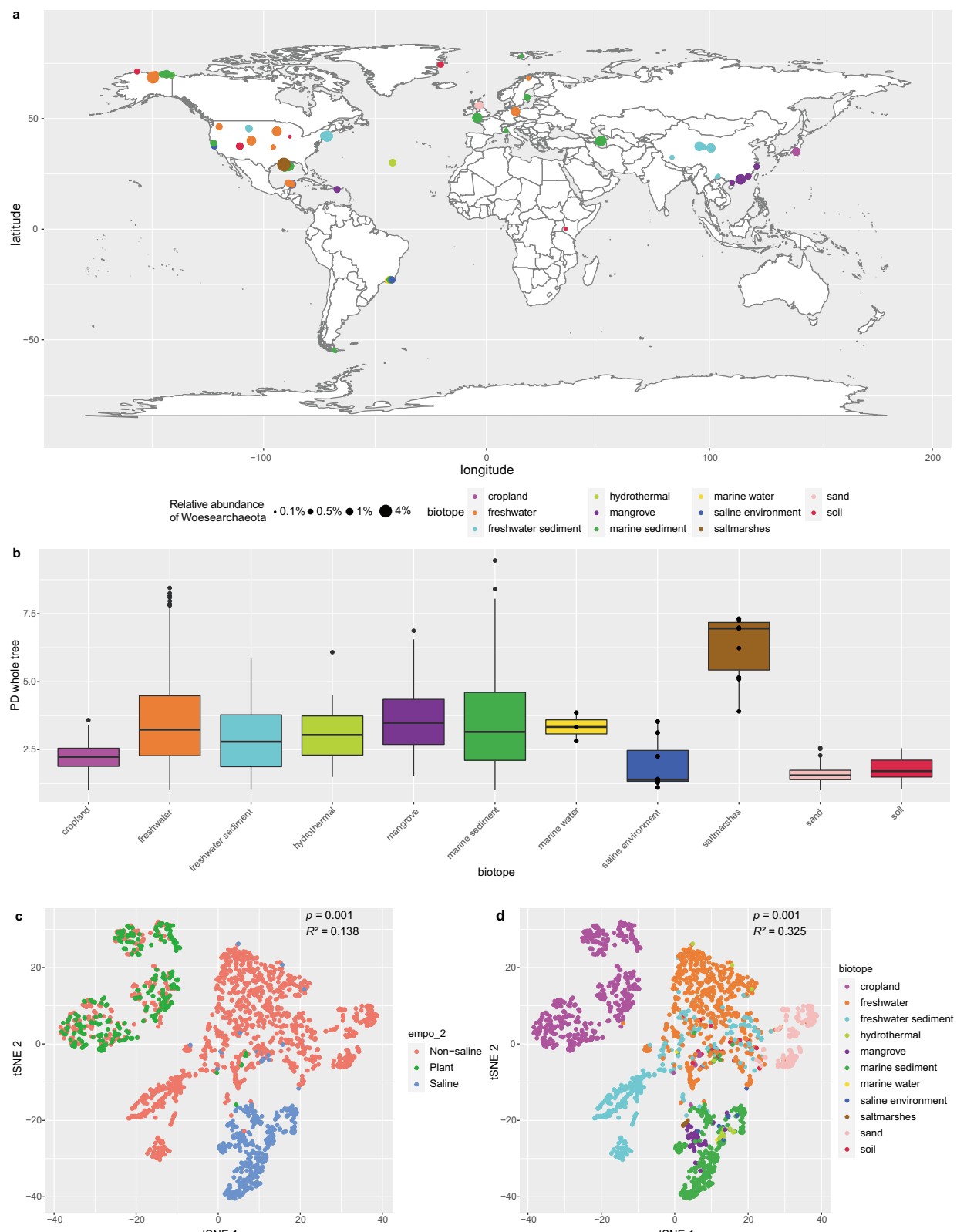

genome size exceeded 1 Mbp. In contrast, members of subgroup A, deeply rooted within Woesearchaeota in the phylogenetic tree, had the smallest median estimated genomic size (0.98 Mbp, Fig. 2; Supplementary Fig. 3). By contrast, the relationship between the phylogenetic group and genomic GC content was not pronounced. The genomic GC content in several subgroups greatly varied internally (Fig. 2; Supplementary Fig. 3). For

example, estimated genome sizes of archaea in subgroup G, range from 0.88 to 1.36 Mbp (average $1.31 \pm 0.33$ Mbp), with the GC content ranging from 30 to 59%.

The 16S rRNA gene sequences identified in the MAGs allow us to link the subgroups defined here with previously described clusters[18] (Fig. 2; Supplementary Fig. 4; Supplementary Table 2). Subgroup A, C, E, H, I, and J contain sequence representatives in

**Fig. 1 Global distribution of Woesearchaeota. a** Global distribution of Woesearchaeota with at least 0.1% relative abundance, based on 2,163 16S rRNA gene amplicon datasets. Note that some libraries have the same coordinate and thus are overlapping. The world map was generated using R package "maps" v3.3.0, in R v3.6.0[97]. **b** Box plot of phylogenetic diversity whole-tree index of Woesearchaeota in 11 biotopes. The median is shown in a thick black bar. The first and third quartile are shown respectively in the lower and upper bound of the box. The whiskers correspond to the 1.5 interquartile range from the bounds. The outlier is marked by dots. Data points from samples with sizes of less than or equal to ten are shown. The number of replicates used for different biotopes are as follows: freshwater: 629, cropland: 590, freshwater sediment: 365, marine sediment: 293, sand: 174, mangrove: 46, soil: 30, hydrothermal: 15, saltmarshes: 10, saline environment: 8, marine water: 3. **c** Beta diversity plots of Woesearchaeota community based on 2163 16S rRNA gene amplicon datasets. Beta diversity was calculated by the unweighted UniFrac metric method and tested with a 999-permutation PERMANOVA test. The dots are colored by empo_2 ontology[41] which includes saline, non-saline, and plant environment. **d** Beta diversity plots of Woesearchaeota community based on 2163 16S rRNA gene amplicon datasets. Beta diversity was calculated by the unweighted UniFrac metric method and tested with a 999-permutation PERMANOVA test. The dots are colored according to the biotope. Source data are provided as a Source Data file.

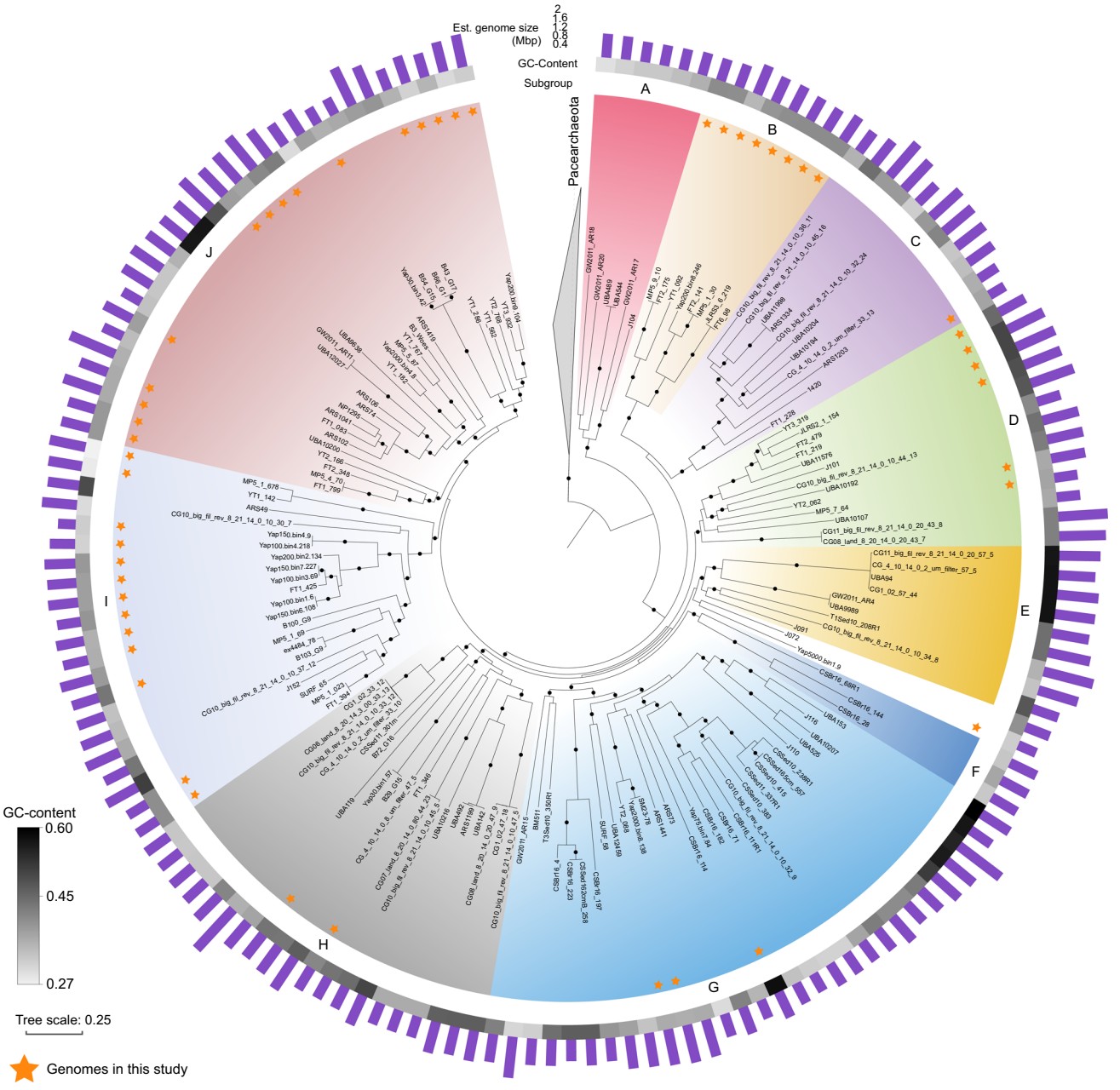

**Fig. 2 Phylogenetic tree of Woesearchaeota.** The maximum likelihood tree was generated based on the concatenated 50% top-ranked orthologs ($n = 38$, 10,896 sites) with IQ-Tree (v2.0.7) using LG + F + R + C60 model. The tree includes 152 Woesearchaeota genomes and is rooted by 30 Pacearchaeota genomes. 49 MAGs generated in this study are marked by the star. Black dots denote nodes with UFBOOT support values ≥95% and SH-aLRT values ≥80%. Scale bar shows the average number of substitutions per site. The minimum number of orthologs in Woesearchaeota MAGs is 21. J072 and Yap5000.bin1.9 were ungrouped as they did not form well-supported clade (UFBOOT < 95% and SH-aLRT <80%). Source data are provided as a Source Data file.

Woese-3, Woese-4, Woese-14b, Woese-14a, Woese-24, and Woese-21a, respectively. In addition, subgroup G has sequence representatives in a monophyletic clade in the 16S rRNA gene tree, including Woese-8, Woese-10, Woese-9, Woese-6, Woese-18, and Woese-20, indicating a large diversity in subgroup G that remains unexplored. Next, we used the linked sequence clusters to probe the approximate distribution of subgroups in the 2163 16S rRNA gene libraries. The analysis showed that subgroup I, J, and G were detected among all biotopes investigated (with >73% occurrence in the studied libraries; Supplementary Data 2), indicating high ecological adaptability of these lineages (Supplementary Fig. 5).

We next inferred the metabolic potential of 152 genomes (≥50% completeness and ≤10% contamination) representing the 10 Woesearchaeota subgroups. We confirmed some previously described metabolic features of Woesearchaeota[16–18], namely: (1) while all genomes appear to lack complete electron-transport chains, including NADH dehydrogenase and complexes II–IV of the oxidative phosphorylation chain, some encode complex V (V-type ATPase) (Fig. 3a; Supplementary Data 3); (2) most genomes encode few components of tricarboxylic acid (TCA) cycle; (3) while the lack of complete glycolytic pathway and phosphofructokinase (pfk) is common, few genomes (n = 14) do encode pfk (Supplementary Data 3). Notably, two genomes (YT1_182 and Yap2000.bin4.8) from subgroup J exhibited the potential for complete glycolysis (Fig. 3b; Supplementary Data 4).

We then used the Carbohydrate-active Enzymes (CAZy) database to assess Woesearchaeota capacity to degrade complex carbon sources. The analysis revealed that 78 genomes, including YT1_182 and Yap2000.bin4.8 from subgroup J (Fig. 3b), encode as least one copy of alpha-amylase (GH57), supporting potential for starch degradation among Woesearchaeota. Most Woesearchaeota members (mainly from subgroups B, C, D, H, I, and J) appear to use pyruvate/2-oxoacid-ferredoxin oxidoreductase (porA/porB) to decarboxylate pyruvate for acetyl-CoA synthesis. By contrast, some members (mainly from subgroups A and J) appear to employ pyruvate dehydrogenase, which is generally found among aerobes (mostly Eukarya and Bacteria). Genes for phosphate acetyltransferase (pta) and acetate kinase (ackA), which catalyze the synthesis of acetate (the Pta-AckA pathway), are present in many subgroups, except for subgroups A, B, C, and I. With the presence of acetyl-CoA synthetase (acsA), pta, and ackA, MAGs such as YT1_182 and Yap2000.bin4.8 may be able to regulate electron flow by consuming or generating acetate (Fig. 3b). However, further fermentation into ethanol might be disabled because the gene for aldehyde dehydrogenase is apparently missing (Fig. 3b). In addition, other than alcohol and acetate, many MAGs were also predicted to encode lactate dehydrogenase (Supplementary Data 3). Apart from glucose, using extracellular DNA as a growth substrate is also possible for Woesearchaeota, as intermediates of nucleotide degradation, such as glycerate-3-phosphate, could be channeled into the second half glycolytic pathway. For example, some members from subgroups A, G, H, and J encode a complete nucleoside degradation pathway, including AMP phosphorylase (deoA), ribose-1,5-bisphosphate isomerase (R15Pi), and ribulose 1,5-bisphosphate carboxylase (rbcL), in which the main form of rbcL among Woesearchaeota represents class III-3b (Supplementary Fig. 6). Many pathways identified herein in Woesearchaeota require the participation of cofactors, such as NAD+/NADH, required for glycolysis, and CoA, needed in the Ack-Pta pathway. Nonetheless, the biosynthetic pathways for cofactors (e.g., cobalamin, CoA, and thiamine) were rarely complete in these genomes (Supplementary Data 3). Collectively, these observations indicate that limited metabolic potential in carbohydrate metabolism is common among Woesearchaeota.

## H₂ metabolism might play roles in the generation of proton-motive forces in subgroup J.

Hydrogenases, the key enzymes in hydrogen metabolism, can employ $H_2$ as reducing power or $H^+$ as oxidants to dissipate excessive reductants in cells. They were commonly found among different Woesearchaeota subgroups and appeared to be important for Woesearchaeota. In the placement of a well-defined electron transport chain (Complex I −IV), hydrogenase could re-oxidize NADH, NADPH and reduced ferredoxin by reducing $H^+$ to molecular hydrogen. Specifically, we identified putative cytosolic [NiFe] hydrogenases (group 3b), which evolve $H_2$ by coupling the oxidation of NADPH and may be reversible in the genomes of subgroup B, D, I, and G (Supplementary Fig. 7). The hydrogenase is also encoded in genomes of multiple diverse bacterial and archaeal phyla (e.g., Proteobacteria and Euryarchaea)[20]. Remarkably, [FeFe] hydrogenases, the most efficient enzymes for catalytic hydrogen turnover that are typically found in bacteria and eukaryotes[21,22], were identified in Woesearchaeota, i.e., subgroup B, E, G, H, and J. Phylogenetic analysis of their catalytic domain indicated that they belong to group A [FeFe] hydrogenase (Fig. 4a). Further examination into the genetic organization of group A [FeFe] hydrogenases encoded in Woesearchaeota showed that most of them are trimeric, containing a catalytic subunit (H-cluster), a nuoF-like, and a nuoE-like gene (Supplementary Fig. 8). The nuoF-like gene encodes a 4Fe−4S cluster binding domain, an electron carrier flavin mononucleotide (FMN)-binding domain, and a soluble ligand-binding domain. Therefore, the trimeric [FeFe] hydrogenases in Woesearchaeota belong to [FeFe] hydrogenase group A3[20,21], which can reversibly bifurcate electrons from $H_2$ to ferredoxin and $NAD^+$. Cysteine residues of Woesearchaeota A3 [FeFe] hydrogenases that bind the metal ions are conserved, suggesting the presence of active sites (Fig. 4c). Phylogenetically, the catalytic subunit of these hydrogenases formed a monophyletic group with some other DPANN archaea (i.e., Micrarchaeota, Aenigmarchaeota, and Nanohaloarchaeota) and are closely related to the hydrogenases in Thermotogae and Bacteroidetes (Supplementary Fig. 9).

Additionally, we identified a potential membrane-bound complex named Rhodobacter nitrogen fixation (Rnf) electron transport complex in four members of subgroup J (i.e., YT1_182, Yap2000.bin4.8, YT1_767, and MP5_5_87). This complex could serve as a respiratory enzyme that couples the oxidation of reduced ferredoxin to reduce $NAD^+$. The free energy of this exergonic reaction could be used to translocate sodium ion or proton out of cells, thereby generating a potential gradient. The putative V-type ATP synthase could then harness the gradient to conserve energy by making ATP. The Rnf complex operon in the four genomes consisted of six subunits (rnfCDGEAB), resembling the organization of the operon in Acetobacterium woodii[23]. Phylogenetic analyses of these six genes indicated that they might have been transferred from bacteria (Supplementary Figs. 10−15). The electrochemical gradient could also be generated by a putative pyrophosphate-energized sodium pump (hppA), which utilizes the energy of pyrophosphate hydrolysis to move $Na^+/H^+$ across the membrane (Fig. 3b; Supplementary Data 4).

The four MAGs in subgroup J also have genes encoding a putative butyryl-CoA dehydrogenase (Bcd) adjacent to genes encoding electron transfer flavoprotein subunits EtfA and EtfB. They may constitute an EtfAB/Bcd complex, which catalyzes short-chain acyl CoA transformation to short-chain trans-2,3-dehydroacyl-CoA, with NADH as the electron donor, ferredoxin as the negative redox potential acceptor, and short-chain acyl CoA as the positive potential acceptor[21]. Phylogenetic analysis indicated Bcd genes in the four MAGs are most related to Firmicutes (Supplementary Fig. 16). However, the absence of the

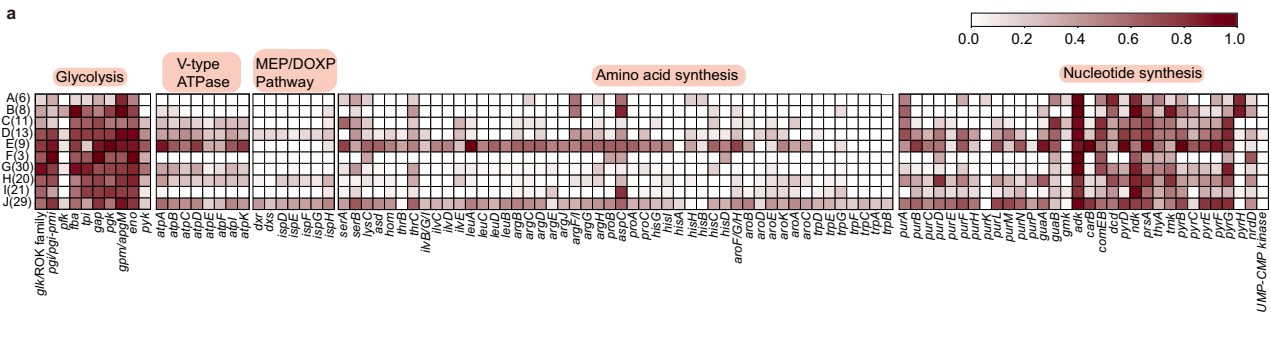

**Fig. 3 Metabolic comparison of Woesearchaeota subgroups. a** Occurrence of a gene of interest in different subgroups of Woesearchaeota. The occurrence was calculated in percent across the total number of genomes included in each subgroup based on the presence/absence table. Raw data are available in Supplementary Data 3. **b** Reconstructed metabolic pathways of two MAGs from subgroup J, YT1_182 (estimated to be 94.61% complete and 1.96% contaminated) and Yap2000.bin4.8 (estimated to be 94.61% complete and 3.92% contaminated). A detailed list of genes encoded by these two MAGs can be found in Supplementary Data 4. Source data are provided as a Source Data file.

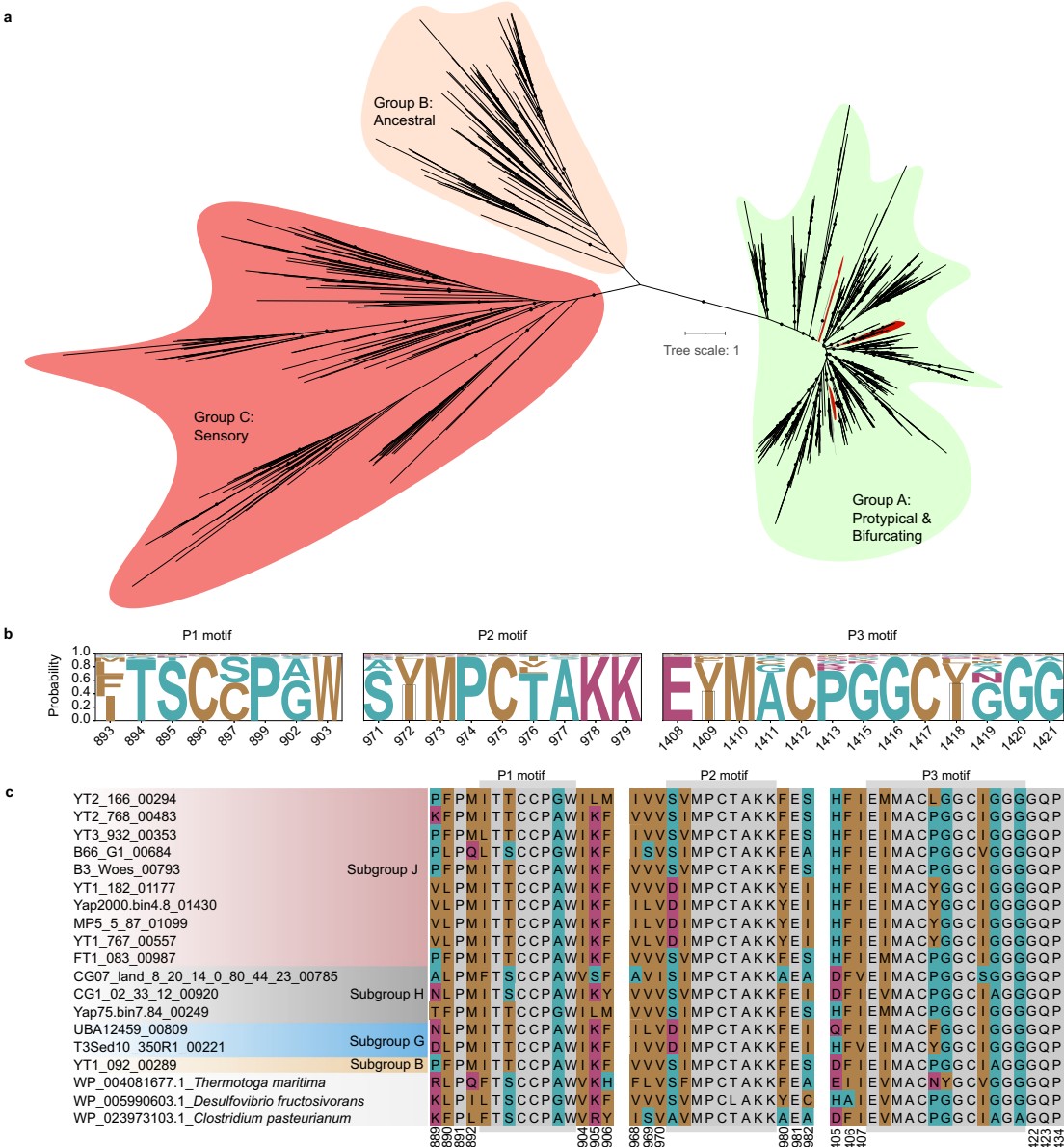

**Fig. 4 Diversity of [FeFe] hydrogenases in Woesearchaeota. a** Phylogenetic analysis of the catalytic domain of [FeFe] hydrogenase sequences ($n = 856$, trimmed alignment length = 356) based on LG + G + C10 model using IQ-Tree (v1.6.8). Black dots denote nodes with UFBOOT values ≥95%. Scale bar shows the average number of substitutions per site. Woesearchaeota sequences are colored in red embedded in the group A sequences. **b** Probability plot of the occurrence of each amino acid of the P1, P2, and P3 motif of [FeFe] hydrogenase group A3. Amino acids are colored based on their hydrophobicity score. Gray box is added to distinguish the letters V and I from Y. Positions of alignment are indicated below the amino acids. **c** Conservation of the P1, P2, and P3 motif of selected amino acid sequences in comparison to *Thermotoga maritima* (WP_004081677.1), *Desulfovibrio fructosivorans* (WP_005990603.1), and *Clostridium pasteurianum* (WP_023973103.1). The difference to the reference sequences is colored by their hydrophobicity score and conserved site is colored in gray. The position of amino acids flanking the P1, P2, and P3 motif in the alignment is indicated at the bottom of the plots. Source data are provided as a Source Data file.

FAD-binding domain in EtfAB renders electron bifurcation unlikely in this complex[24].

To summarize, our analyses indicate that hydrogenases may play important roles in the metabolism of Woesearchaeota. In subgroup J, like YT1_182, they might employ hydrogenase and Rnf complex and other electron bifurcation complex to balance the reducing pool (NADH and ferredoxin), especially reducing equivalents from the glycolytic pathway. Energy could be conserved by substrate-level phosphorylation and the coupling of the Rnf complex with ATP synthase. Therefore, based on the metabolic analysis, we predict that members of subgroup J may have a heterotrophic lifestyle with fermentative metabolism.

**Biosynthetic capacity differentiates subgroup J from other Woesearchaeota.** Compared with other Woesearchaeota subgroups, subgroup D and J Woesearchaeota appeared to harbor expanded gene inventories related to the biosynthesis of amino acids, nucleotides, and isoprenoids (Fig. 3a). In terms of amino acid biosynthesis, members of subgroups D and J harbor various genes for the de novo synthesis of arginine, serine, proline, leucine, and threonine (Fig. 3a; Supplementary Data 3). Members of the subgroup J were also found to encode genes required to synthesize tryptophan (Fig. 3a). Further, some members of the subgroup J (e.g., YT1_182 and Yap2000.bin4.8) encode genes necessary for the de novo synthesis of proline, valine, and leucine.

Still, the pathway for the synthesis of complex amino acids, such as aromatic amino acids, is incomplete (Fig. 3b). Some amino acids like arginine might be translocated into cells through an amino acid/polyamine transporter (Fig. 3b).

The difference in the occurrence of genes for the purine and pyrimidine biosynthesis between subgroups was not as striking as that in the occurrence of genes for amino acid biosynthesis (Fig. 3a). Nonetheless, of note, genomes from subgroups D and J harbor many genes for nucleotide biosynthesis. For example, we recovered nearly complete pathways for the purine and pyrimidine synthesis from YT1_182 and Yap2000.bin4.8 (Fig. 3b; Supplementary Data 4).

Furthermore, only the bacterial MEP pathway for the synthesis of isoprenoid precursors (isopentenyl diphosphate and dimethylallyl diphosphate) existed in Woesearchaeota genomes. More interestingly, the pathway appeared to be confined to subgroups F and J, and this pathway was present in nearly half of the members of subgroup J, including YT1_182 and Yap2000.bin4.8 (Fig. 3). Most members of subgroup J have a fused gene encoding 2-C-methyl-D-erythritol 4-phosphate cytidylyltransferase (ispD) and 2-C-methyl-D-erythritol 2,4-cyclodiphosphate synthase (ispF) domains. Phylogenetic analyses of genes in the pathway indicated that they are closely related to sequences of different bacterial organisms (Supplementary Figs. 17–22). For example, ispD/F is most related to Candidatus Nealsonbacteria (Supplementary Fig. 19), while 1-deoxy-D-xylulose 5-phosphate reductoisomerase (dxr) and 4-hydroxy-3-methylbut-2-en-1-yl diphosphate synthase (ispG) of subgroup J are most related to Firmicutes (Supplementary Figs. 18, 21). Nevertheless, these archaea might be unable to synthesize membrane lipids because of the absence of two crucial genes, namely geranylgeranylglyceryl phosphate synthase and digeranylgeranylglyceryl phosphate synthase.

**Evolutionary history of subgroup J.** Given members of subgroup J have more genes involved in isoprenoid and amino-acid biosynthesis and some may have a distinct lifestyle (i.e., heterotrophic fermentative metabolism) from other Woesearchaeota, we sought to reconstruct the gene family evolutionary history of subgroup J, hoping to gain insights into the lifestyle transformation. As data completeness could affect the accuracy of ancestral reconstruction, we selected 47 Woesearchaeota genomes for further analysis according to the criterion (>79% completeness and <5% contamination). This criterion was set according to the quality metric of the first circular complete MAG reconstructed in Castelle et al.[6], Woesearchaeota AR20, whose completeness was estimated to be 79.17% in this study. On the other hand, we opted to use the gene-tree-aware approach implemented by ALE which incorporates a probabilistic method to account for the estimated missing fraction of genomes[25]. Taxonomic sampling also affects the accuracy of ancestral reconstruction. Here, 47 representative MAGs were selected according to their quality metrics. They covered nine subgroups except for subgroup F. We then clustered 65,396 genes from 47 Woesearchaeota genomes and 5 Pacearchaeota genomes into 4,562 gene families. Gene tree samples were inferred for the 2,320 gene families with ≥4 sequences. They were probabilistically reconciled with the maximum likelihood tree inferred for the 52 genomes. Phylogenetic placement of 47 Woesearchaeota genomes in the tree was in good agreement with their positions in the tree used for subgroup assignment. It was also well supported, with most placements achieving maximal support (Supplementary Fig. 23). We chose to report ancestral events and gene copies numbers with a minimum threshold of 0.3. This threshold is relaxed and believed to be able to detect ancestral events blurred by noises from alignment, tree

reconstruction, and reconciliation[26]. The ancestral reconstruction result is summarized in Fig. 5a.

The analysis predicted that a recent major expansion of subgroup J genomes occurred recently, multiple times (node 87 to node 82, node 82 to node 79, node 82 to node 73, and node 79 to node 74) (Fig. 5a). Nearly half of the genes predicted to gain at nodes 82 and 79 are related to metabolism (Fig. 5b). Further, at node 82, a large proportion of acquired genes was related to amino acid transport and metabolism (approximately 16%), and nucleotide transport and metabolism (approximately 10%). According to the minimum frequency threshold, the predicted gene content gains mainly covered 110 originations events (de novo genes or transfers from unsampled genomes, 43% of gains) and 83 intra-transfers events (transfers from sampled genomes, 32% of gains). At node 79, the acquired genes were mainly related to amino acid transport and metabolism (approximately 11%), and energy production and conversion (approximately 13%) and mainly occurred by 38 originations (15% of the gains) and 84 intra-transfers (33% of the gains). Interestingly, genes involved in isoprenoid synthesis (e.g., dxs, ispE, ispD/F, ispG, and ispH), amino acid biosynthesis (e.g., leuA, hisG, and trpG), and nucleotide synthesis (e.g., purE and pyrE) were predicted to be gained at node 82. Our analyses suggest that half of those apparent gene gains occurred through origination events, possibly by acquisition from bacteria (Supplementary Data 5; Supplementary Figs. 17–21; Supplementary Figs. 24–38). Genes involved in energy production and conversion including Rnf complex and EtfAB/Bcd complex were apparently acquired at node 79 through origination events, possibly transferred from bacteria as well (Supplementary Figs. 10–16). Furthermore, approximately 10 and 7% of genes gained at node 73 were related to amino acid transport and metabolism, and nucleotide transport and metabolism, respectively (Supplementary Data 5). These apparent gain events may have contributed to the enhanced metabolic potential of subgroup J.

## Discussion

Woesearchaeota are one of the most ubiquitously distributed lineages within the DPANN superphylum[18,27]. In the current study, we retrieved the Woesearchaeota 16S rRNA gene sequences from different biotopes, to understand the global distribution patterns of Woesearchaeota, and collated 103 Woesearchaeota genomes with 49 new ones reconstructed in this study to gain insights into the metabolism and evolutionary history of Woesearchaeota diversification.

We here first updated the previously published 16S rRNA gene-based division of Woesearchaeota subgroups[18], by inclusion and robust phylogenomic inference from newly reconstructed genomes (see Supplementary Note 1). We anticipated that the newly defined Woesearchaeota subgroups A–J would facilitate understanding of their global distribution pattern. We showed that Woesearchaeota abundance is relatively low in natural environments, although their roles in maintaining the community stability, considering their high diversity, might be important[28,29]. Based on the phylogenetic diversity index analysis, Woesearchaeota are more abundant and more diverse in saltmarshes and mangroves than in other biotopes. Sand and soil appear to be the least preferred habitats of Woesearchaeota. Finally, the approximate link between genome subgroups and 16S rRNA gene sequence clusters indicated subgroup G, I, and J may have high ecological adaptability. However, more 16S rRNA gene sequences from genomes are needed to accurately investigate subgroup distribution in the environment.

Despite the wide distribution across 11 distinct biotopes, Woesearchaeota appear to share some common metabolic

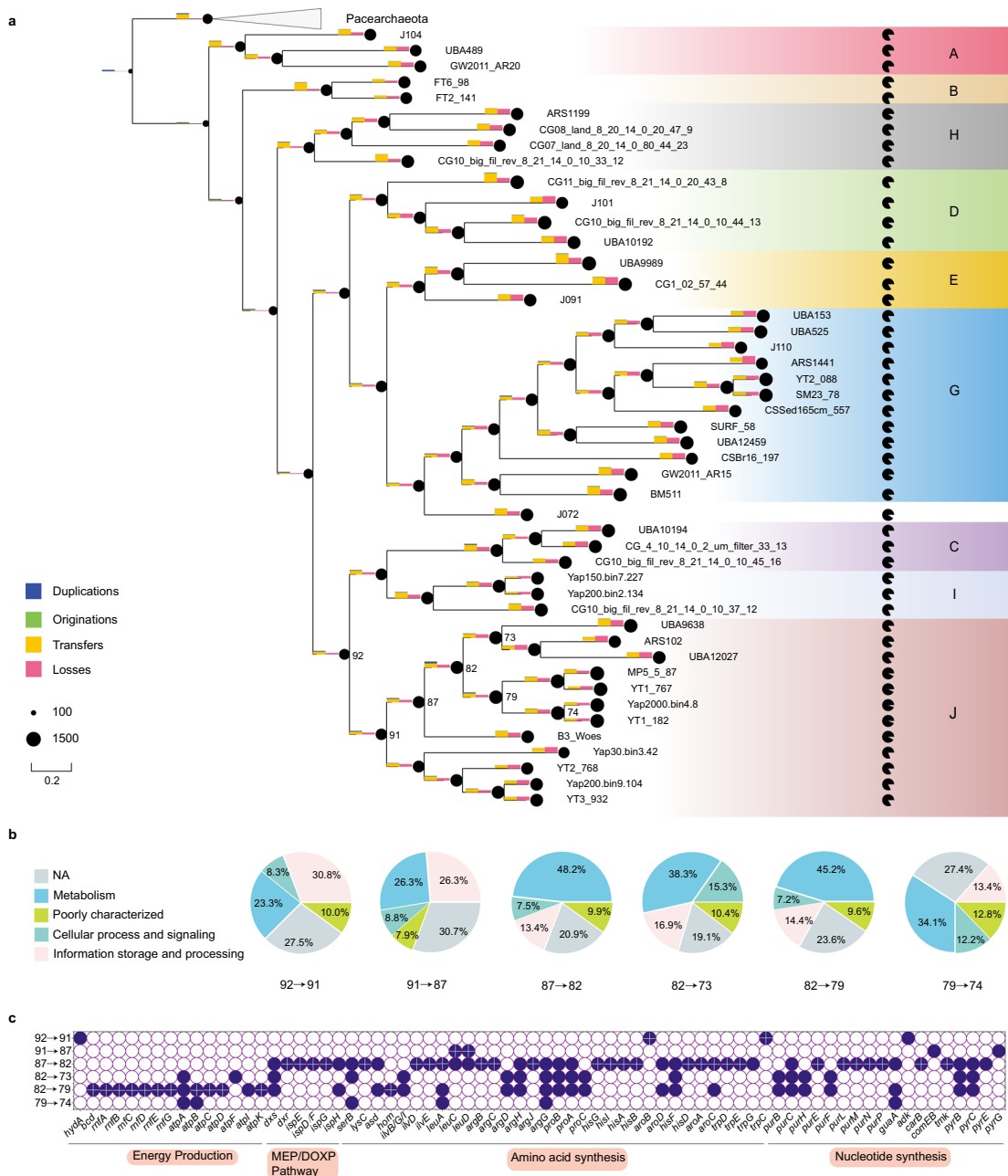

**Fig. 5 Evolution of gene family for subgroups in Woesearchaeota. a** Summary of ancestral events in the evolution of gene families for Woesearchaeota. The phylogenetic tree includes 47 Woesearchaeota MAGs representing major subgroups and is rooted by 5 Pacearchaeota MAGs. The full tree is shown in Supplementary Fig. 23. The Pacman chart shows the estimated completeness of Woesearchaeota MAGs. The area of black circles before nodes and at tips indicates the number of their inferred gene families. The height of bars along the branch corresponds to the number of duplications, originations, transfers, and losses (the height in the legends corresponds to 1000 events). For visualizing purpose, some branches were extended with dashed lines. **b** The pie charts represent functions of gained genes at nodes of interest-based on COG category information. NA means no annotations. Gene gain events were counted if the inferred frequency is over 0.3. **c** Genes of interests predicted to gain at nodes of interests. Events projected as origination are marked with a cross. Note that events are counted if the inferred frequency is greater than 0.3. Raw data can be found in Supplementary Data 5. Source data are provided as a Source Data file.

abilities, including the lack of complete TCA cycle and electron transport chains (complexes I−IV), and the minimal capacity to synthesize biomolecules, such as nucleotides, precursors for isoprenoids, amino acids, and vitamins. Further, a complete nonoxidative phase of the pentose phosphate pathway and some genes involved in fermentation, such as *ldh*, were recovered from most genomes. Hence, being consistent with the previous analyses[6,17,18], these features highlight a conspicuous metabolic

deficiency of Woesearchaeota, and indicate that most of these archaea might mainly lead an anaerobic and parasitic/fermentation-based lifestyle. How these organisms acquire nutrients and essential building blocks remains unclear because of the lack of pure cultures. These archaea may be intimately associated with other microorganisms, akin to the relationship between *N. equitans* and *I. hospitalis*[8], so as to obtain necessary cellular metabolites, such as amino acids, nucleotides, and lipids. The

diversity-generating retroelements abounding in DPANN (highly represented in Woesearchaeota) may also aid the adaption to such a lifestyle[30].

Despite the above unifying traits, the metabolic potentials of Woesearchaeota subgroups vary pronouncedly. Specifically, subgroup J microbes appear to harbor genes involved in the biosynthesis of amino acids and nucleotides more frequently than the other subgroups, suggesting a relatively greater biosynthetic capacity. Interestingly, the MEP pathway, prevalent in bacteria that synthesize isoprenoid precursors, appears to be specific to subgroup J. These differences highlight the notion that Woesearchaeota are diversified organisms, as also confirmed by the genomic size variation, and phylogenetic analysis based on both, the 16S rRNA gene and single-copy orthologs. More surprisingly, two nearly complete MAGs from subgroup J (Yap2000.bin4.8 and YT1_182) encode the complete glycolysis pathway and an amylase (GH57), suggesting their full potential to metabolize starch. In the glycolytic pathway, two NADH and four reduced ferredoxins are generated respectively in the conversion of glyceraldehyde 3-phosphate to 3-phospho-D-glycerol phosphate via glyceraldehyde-3-phosphate dehydrogenase and in the pyruvate decarboxylation by pyruvate/2-oxoacid-ferredoxin oxidoreductase. These two reducing agents could be synergistically utilized by [FeFe] hydrogenase driving the evolution of $H_2$. Reduced ferredoxins could also fuel the translocation of sodium ion or proton across the membrane by Rnf complex, generating potential gradient and NADH. In EtfAB/Bcd complex, NADH donates electrons to reduce ferredoxin. Therefore, it is likely that the balance of the reducing pool is maintained by the glycolytic pathway, [FeFe] hydrogenase, Rnf complex, and EtfAB/Bcd complex in these Woesearchaeota. These results indicate that some subgroup J members, if not all, may be capable of anaerobic heterotrophy with fermentative metabolism. To some degree, their lifestyle may have a resemblance with some fermentative bacterial organisms like *Thermotoga maritima*[31,32] and *Clostridium thermocellum*[33]. According to previous studies[16,17], most CAZymes of DPANN archaea are extracellular. It is also possible that subgroup J members secrete some CAZymes and peptidases into the surrounding medium to decompose starch or other organic matter derived from dead cells or exuded by living cells. Therefore, subgroup J might be able to associate themselves with particles. However, fluorescent in situ hybridization, isotopic labeling, or pure culture experiments are essential to confirm the actual state or lifestyle of these organisms[34,35]. [FeFe] hydrogenases are capable of catalyzing $H_2$ formation with efficiency hitherto unparalleled[22]. Ecologically, members of subgroup J might be important in anoxic environments like hydrothermal vents and marine sediments where they occur more frequently (Supplementary Fig. 5f) and could benefit other microbes like $H_2$-utilizing methanogenic archaea[18] by producing $H_2$. Therefore, the above observations suggest that Woesearchaeota may impact the carbon and hydrogen cycle[6].

Whether DPANN archaea form a clan is still debated[36,37], as some DPANN lineages exhibit high rates of sequence evolution, making them vulnerable to long-branch attraction. Some recent reports support the clanhood of DPANN archaea[36,37]. Further, inference of the ancestral metabolism of DPANN archaea revealed incomplete glycolysis pathway and TCA cycle in their common ancestor[37]. Woesearchaeota are placed together with Nanoarchaeota, Nanohaloarchaeota, Parvarchaeota, Pacearchaeota, Aenigmarchaeota, and Huberarchaeota in most reports[6,27,37]. Owing to their reduced genomes, DPANN archaea are thought to be symbionts or parasites of other prokaryotes[6,7]. Indeed, the host dependence of some of these organisms has been substantiated. For example, *N. equitans*, the first DPANN archaeon to be characterized, is obligately dependent on *I.*

*hospitalis*[8], and *Candidatus* Nanohaloarchaeum antarcticus requires *Halorubrum lacusprofundi* as a host for growth[10], suggesting that symbiotic or parasitic lifestyle may be a common feature of these DPANN lineages. Nevertheless, the hypothetical lifestyle of DPANN has been recently challenged in a study that proposed that most DPANN archaea do not lead a symbiotic lifestyle in subsurface environments[11].

The presence of a complete glycolysis pathway and extensive repertoires of genes for amino acid and nucleotide biosynthesis as well as hydrogenase in subgroup J highlights the metabolic flexibility of these microbes. Subgroup J archaea first experienced gains of genes related to the transport and metabolism of amino acids and nucleotides and then genes of energy production and conversion. These two steps perhaps enable a reduced dependence on a host or shift to another life strategy that does not require cell–cell association to obtain exogenous cellular components, unlike many other auxotrophic microorganisms[38]. The acquisition of these genes has putatively enhanced the biosynthetic capacity and energy production of these organisms. However, currently, the underpinning drivers are unclear. The acquisition of genes may have been induced by symbiont switching or an alternative resource acquisition strategy[39]. Bacterial organisms, most likely, play major roles in the origination of additional metabolic traits in subgroup J. The acquisition of many genes related to isoprenoid and amino acid biosynthesis is predicted as origination events and likely from bacterial donors. Likewise, genes encoding protein complex vital for energy production like Rnf complex are also projected as originations and possibly transferred from bacteria. Although, *pfk*, *pta,* and *ackA*, important genes in carbohydrate metabolism, are not inferred as gain through originations at the nodes of subgroup J, phylogenetic analyses indicated that they are likely transferred from bacteria as well (Supplementary Figs. 39−41). For example, *pfk* of subgroup J are phylogenetically closely related to those of *Candidatus* Abyssubacteria (Supplementary Fig. 39). This is consistent with previous analysis which showed genes related to carbohydrate metabolism are prone to lateral gene transfer[37].

## Methods

**Processing of 16S rRNA gene amplicon data**. Raw sequence data were retrieved from sediment samples (78 samples)[40] collected along the southeast coast of China and from the EMP[41], based on in silico binding analysis of primers 515F and 806R[41]. These data were then combined and trimmed using Sickle (v1.33; https://github.com/najoshi/sickle) and the "-q 25" setting to control sequence quality. Sample metadata were subsequently retrieved from a reference paper[40] and the EMP website (https://earthmicrobiome.org/protocols-and-standards/metadata-guide/). To remove chimeric sequences, data that passed quality control were searched using VSEARCH[42] (v2.13.3) against the Greengenes database (v13_8)[43]. The reads were clustered into OTUs at the threshold of 97% sequence identity. Taxonomy was assigned using the "assign_taxonomy.py" function in QIIME[44] (v2018.11) against the custom database composed of SILVA SSU 132 database (https://www.arb-silva.de/documentation/release-132) and an in-house Woesearchaeota archaea database. Finally, 2163 16S rRNA gene libraries were selected, using the following criteria: minimum sequencing depth of 30,000 and minimum relative abundance of Woesearchaeota in the libraries of 0.1%. OTUs were aligned with "align_seqs.py" using the default setting. The OTU table was rarefied to 30,000. Alpha and beta diversity were calculated using "alpha_diveristy.py" and "beta_diversity.py" respectively. Metadata of libraries used in the final analysis could be found in Supplementary Data 2.

**Sample collection and sequencing**. Six YT sediment samples were collected on November 15, 2018 from a seagrass bed and a neighboring site not covered by seagrass at the Rongcheng Swan Nature Reserve (Rongcheng) as described in Liu et al.[45]. Samples were taken at depths of 0−2, 21−26, and 41−46 cm, sealed in sterile plastic bags, and transported to the laboratory in a pre-cooled container. Total DNA was extracted from 10 g of each sample using PowerSoil DNA Isolation kit (QIAGEN, Hilden, Germany), according to the manufacturer's protocol. The extracted DNA was then sequenced with Illumina HiSeq2500 (San Diego, California, U.S.) PE150 at Novogene.

Five FT sediment samples from Futian Nature Reserve (Shenzhen) were collected on April 17, 2017 as described for YT samples at depths of 0−2, 6−8,

12–14, 20–22, and 28–30 cm. For each sample, 5 g sediments were used to extract DNA using DNeasy PowerSoil kit (Qiagen) as per the manufacturer's instructions. Sequencing data were generated with Illumina HiSeq2000 (Illumina) PE150 at Novogene.

Five sediment samples were obtained on September 12, 2014 from Mai Po Nature Reserve (Hong Kong). Three were taken at a mangrove-forest-covered site at depths of 0–2, 10–15, and 20–25 cm, and two at an intertidal mudflat with depths of 0–5 and 13–16 cm. Samples were sent back to the laboratory as described for the YT samples. DNA extraction for each sample was performed with 5 g wet sediment using PowerSoil DNA Isolation Kit (MO BIO) following the manufacturer's protocol. Nucleic acids were sequenced with Illumina HiSeq2500 (San Diego, California, U.S.) PE150 at Novogene.

The seawater samples were collected during the 37th Dayang cruise at Yap trench region using CTD SBE911plus (Sea-Bird Electronics). 8l seawater of each sample was filtered with a 0.22 μm-mesh membrane filter on board. The membrane was cut into about 0.2 cm² pieces and used to extract DNA with PowerSoil DNA Isolation Kit (MO BIO) following the manufacturer's recommendations. In accordance with the manufacturer's protocol, the DNA of each sample was first amplified in five separate reactions using REPLI-g Single Cell Kits (Qiagen) and the products were then pooled and purified using QIAamp DNA Mini Kit (Qiagen). Milli-Q water (18.2 MΩ; Millipore) was used as parallel blank controls in the filtration, DNA extraction, and amplification processes. Nucleic acids were sequenced with HiSeq X Ten (Illumina) PE150.

Two sediments samples were collected from Jiulong River estuary using a grabber during a cruise on November 28, 2018. The sediments were immediately sealed in 50 ml tubes (Falcon), frozen in liquid nitrogen on board and kept at −80 °C. DNA extraction was performed with PowerMax soil kit (Qiagen) as per the manufacturer's instructions. The DNA fragments were purified, end-repaired, poly(A) tailed, and ligated with Illumina-compatible adapters prior to sequencing as described in Zou et al.[46]. Shotgun sequencing was performed using the Illumina HiSeq 2000 (San Diego, California, U.S.) PE150.

### Genome-resolved binning

*YT, FT, and MP5 metagenomes.* The assembly and binning of the YT, FT, and MP5 sets of metagenomes were performed using the same method. The raw shotgun metagenomic sequencing reads were trimmed using the "read_qc" module in metaWRAP pipeline[47] (v1.2.4) with the setting "—skip-bmtagger". High-quality reads were then merged and co-assembled using MEGAHIT[48] (v1.1.3) with default settings. The binning analyses were performed eight times with eight different combinations of specificity and sensitivity parameters ('—maxP 60 or 95' AND '—minS 60 or 95' AND '—maxEdges 200 or 500') with MetaBAT2[49] (v2.12.1) at a minimum sequence length of 2000 bp. Finally, the bins were screened using the Das-Tool[50] (v1.0) program to obtain high-quality and high-completeness bins.

*JLR and Yap metagenomes.* For JLR metagenomes, raw sequencing reads were dereplicated at 100% identity and trimmed with Sickle (v1.33, https://github.com/najoshi/sickle) using default settings. The trimmed reads were assembled using IDBA-UD[51] (v1.1.1) with the parameters: '-mink 65, -maxk 145, -step 10'. The binning processes were performed with MetaBAT2[49] 12 times, with 12 combinations of specificity and sensitivity parameters ('—m 1500, 2000, or 2500' AND '—maxP 85 or 90' AND '—minS 80, 85, and 90') for further refinement. The binning results were pooled and screened using Das-Tool[50] (v1.0). For Yap metagenome, raw metagenomic shotgun sequencing reads were trimmed with Trimmomatic[52] (v0.38). Assembly and binning analyses were performed as described for JLR metagenome for each Yap metagenome.

*Genome selection, and refinement of the completeness and contamination estimates.* Woesearchaeota genomes that were publicly available prior to November 8, 2019, were obtained from the NCBI[53], IMG[54], and GTDB[55] databases. Predefined marker sets for the archaeal domain provided by CheckM (v1.0.12) were used to estimate the completeness and contamination[56], considering the small Woesearchaeota genome size and the phylogenetic placement of DPANN archaea at the base of the archaeal tree in a recent report[37]. To identify maker genes that are absent across 152 Woesearchaeota genomes, the "checkm qa—out_format 4" function was used. This resulted in the exclusion of five marker genes (TIGR01213, PF04019.7, TIGR00432, PF06026.9, and PF01922.12). The final set of 143 marker genes was used to determine the approximate completeness and degree of contamination. Genomes from the public databases and those generated in the current study (see Supplementary Data 1 for details) with an estimated completeness ≥50% and contamination ≤10% were retained.

*Gene calls and functional annotations.* Proteins were predicted using prodigal (v2.6.3) embedded in prokka (v1.13) and the "—kingdom Archaea—metagenome" options[57,58]. All predicted proteins were subjected to a uniform annotation protocol. Specifically, InterProScan (v5.38-76.0, client version)[59] was used to classify protein functions, with applications including CDD[60], Pfam[61], SMART[62], and TIGRFAM[63] enabled. Protein functions were also annotated using eggNOG-mapper (v2) [using the first KO hit and cluster of orthologous groups (COG) hit], based on eggNOG (v5.0) clusters[64]. Additionally, hmmsearch (v3.1b2; settings: -E

1e–4)[65] and the Carbohydrate-active enzymes (CAZymes) database were used to identify carbohydrate-active genes (downloaded from dbCAN2 in July 2019)[66] in Woesearchaeota genomes. Peptidases were identified using DIAMOND BLASTP (v0.9.24; settings: -k 1 -e 1e-10—query-cover 80—id 50)[67] against the MEROPS database (release 12.0)[68]. Transporters were identified using DIAMOND BLASTP (v0.9.24; settings: -e 1e-4) against the Transporter Classification Database (downloaded in November 2020)[69]. Hydrogenases were classified by using HydDB (https://services.birc.au.dk/hyddb/)[70]. The annotations were then manually inspected, and are summarized in Supplementary Data 3.

*Selection of orthologs for phylogenetic analysis of Woesearchaeota.* To enable accurate classification of Woesearchaeota genomes based on phylogeny, we first identified an initial set of 109 orthologues (OGs) present in 85% of the genomes (>79% complete and <5% contaminated) by running the OMA standalone algorithm (v2.4.1)[71]. This set of proteins were annotated with arCOGs hmm models from eggNOG (v5.0)[64] and TIGRFAM (v15.0) database with hmmsearch (v3.1b2)[65] and the best hit for each protein was selected based on the lowest e-value and the highest bit score (see Supplementary Data 6). Based on the annotations, we excluded 12 OGs that are affected by lateral gene transfer (LGT) or have complex gene history according to previous publications[36,72]. For the remaining OGs, whenever available, we extracted HMM profiles from arCOG and TIGRFAM databases and if not present in the databases, profiles were generated with HMMbuild[65]. To minimize the risk of detecting distant paralogs in the reference genomes, these profiles were combined with available TIGRFAM profiles (v15.0)[63], which resulted into 4521 HMM profiles, to query against 152 Woesearchaeota genomes investigated in this study, 364 archaeal (excluding 39 Woesearchaeota genomes) and 3020 bacterial backbone datasets[73] in Dombrowski et al.[36] with hmmsearch (v3.1b2; settings: -E 1e-5)[65].

Afterwards, archaeal and bacterial homologs of the 97 OGs were extracted by the best hit according to e-value and bit-score. OGs that were not single-copy except Woesearchaeota or found in too few genomes were excluded (see Supplementary Data 6). The OGs of interests were then aligned with MAFFT-LINSI (v7.471)[74] and trimmed with BMGE (v1.12; settings: -t AA -m BLOSUM30 -h 0.55)[75]. Single protein trees containing bacterial homologs were first inferred with FastTree (v2.1.10; settings: -lg -gamma)[76] and if they yielded a paraphyletic archaeal lineage and contained signs of LGT, we excluded them from further analysis (see Supplementary Data 6). Next, we inferred the single protein trees for the 74 OGs with IQ-Tree (v1.6.8; settings: -bb 1000 -alrt 1000 -m LG+G)[77] for assessment of suitability for concatenation using the ranking scheme described in Dombrowski et al.[36]. Finally, the top 50% top-ranking single protein trees were manually inspected for signs of contamination and paralogues which were then removed from the single protein sequences. After manual cleaning, we selected sequences affiliated with Woesearchaeota and Pacearchaeota. They were aligned using MAFFT-LINSI[74] and trimmed with BMGE (v1.1.2; settings: -t AA -m BLOSUM30 -h 0.55)[75]. Sequences were then concatenated using a custom script[78] and a maximum-likelihood tree was inferred using IQ-Tree (v2.0.7; settings: -m LG +F+R+C60 -B 1000 -alrt 1000)[79]. The tree was then rooted using Pacearchaeota as an outgroup. The final phylogenetic tree was visualized with iTOL v4 web server[80].

To delineate Woesearchaeota subgroups, we converted the maximum likelihood tree based on the top 50% ranking proteins to an ultrametric one and calculated the relative evolutionary divergence (RED)[81] of each node in the tree using Pacearchaeota as an outgroup. A subgroup was defined when its branch length in the ultrametric tree and RED value exceeded 0.4, except for subgroup A (RED value of 0.3), and its monophyly was well supported (UFBOOT ≥ 95% and SH-aLRT ≥80%) (See Supplementary Fig. 42).

*Inference of proteome content changes across evolutionary history.* 47 Woesearchaeota MAGs from each subgroup, except subgroup F, were selected according to the quality metric (>79% estimated completeness and <5% estimated contamination). The standard was set according to the quality metric of GW2011_AR20, the first published circular MAG of Woesearchaeota[6]. Combing with five Pacearchaeota MAGs (estimated completeness ≥72% and estimated contamination ≤5%), proteins were clustered with the Orthofinder (v2.2.6; settings: -I 1.5)[82], resulting in 4562 protein families, and families with <4 sequences were removed. For each family, a medoid sequence (the sequence with the shortest sum genetic distance to all other sequences) was picked by calculation under the BLOSUM62 substitution matrix. Medoids were annotated against the KEGG KOfam (ver. 2020-06-07) using KofamScan (v1.3.0)[83], against the TIGRFAM (v15.0) database using InterProScan (v5.38-76.0)[59] and against eggNOG (v5.0) database using eggNOG-mapper (v2)[64]. All sequences of each remaining family were aligned with MAFFT-LINSI[74], followed by trimming with ClipKIT (v0.1; settings: -m medium)[84], and single protein family trees were inferred with IQ-Tree (v1.6.8; settings: -m LG+G -bb 1000 -wbtl)[77]. A maximum-likelihood tree of these 52 MAGs was inferred with IQ-Tree (v2.0.7; settings: -m LG+F+R+C60 -B 1000 -alrt 1000)[79] based on the carefully selected orthologs. Using ALEml_undated in ALE 1.0 package[25], UFBOOT trees of the 2320 single protein families were reconciled against the maximum-likelihood tree to infer the frequency of losses, duplications, transfers (gene transfers within the sampled genomes), and originations (gene transfers from unsampled genomes or de novo genes) across each branch of the tree, and the frequency of copies at

each node. The frequency value represents the probability/support of the events. The missing fractions of each MAG were accounted for by the probabilistic method implemented in ALE[25]. ALE outputs were parsed using scripts from a recent publication[85]. We reported events with a frequency threshold of 0.3. This threshold is relaxed and necessary, considering that the noises from alignment and tree reconstructions could diminish the signal of many true events[26]. Gene copies and events were summarized and visualized along the branches of the tree with ETE Toolkit (v3.1.2)[86].

*Phylogenetic analysis of [FeFe] hydrogenase.* Sequences for classifying [FeFe] hydrogenase of Woesearchaeota were retrieved from HydDB[70]. To reduce sampling size, sequences affiliated with [FeFe] hydrogenase groups A, B, and C were dereplicated using Cd-hit (v4.8.1)[87] with a sequence identity of 0.9, 0.65, and 0.65 respectively prior to the addition of [FeFe] hydrogenase catalytic subunit of Woesearchaeota. Catalytic domains of [FeFe] hydrogenases were identified according to Pfam (v32.0)[61] annotations. Amino acid sequences of catalytic domains were aligned with MAFFT-LINSI[74] and trimmed with ClipKIT (v0.1; settings: -g 0.9)[84]. Using the trimmed alignments, a maximum-likelihood tree was inferred based on LG+C10+G model using IQ-Tree (v1.6.8)[77] with 1000 ultrafast bootstraps[88]. Note that group A [FeFe] hydrogenase could not be reliably subdivided by phylogeny and are classified into subtypes based on their genetic organizations[20]. The genetic organizations of all [FeFe] hydrogenase in Woesearchaeota MAGs were visualized with DNA Features Viewer (v3.0.3) in Python 3.7.0[89]. The logo of P1, P2, and P3 motifs of the H-cluster (catalytic domains) was generated with logomaker (v0.8)[90]. The conservation sites of group A3 [FeFe] hydrogenase in Woesearchaeota were compared in reference to *Thermotoga maritima*, *Desulfovibrio fructosivorans* and *Clostridium pasteurianum*.

To explore the closet phylogenetic relatives of [FeFe] hydrogenase catalytic subunit, Woesearchaeota sequences were queried against Genebank non-redundant database (nr, downloaded March, 2020)[53] with DIAMOND BLASTP (v0.9.24)[67]. The top 500 hits of each sequence were retrieved and deduplicated. Combing with Woesearchaeota sequences, they were aligned with MAFFT-LINSI[74] (alignments with <1000 sequences), proceed with ClipKIT trimming (v0.1; settings: -m medium)[84]. Initial trees were generated with FastTree (v2.1.10; settings: -lg -gamma)[76]. To further shrink the sequence dataset, we manually inspected the trees and selected sequence representatives for well-supported clades. This process was reiterated several times till final sequence dataset contains less than 500 sequences. The selected sequences were then re-aligned with MAFFT-LINSI[74] and trimmed with ClipKIT (v0.1; settings: -m medium)[84]. Phylogenetic analysis was performed with IQ-Tree (v1.6.8; settings: -bb 1000 -alrt 1000 -m LG+G +C20)[77]. Phylogenetic trees were visualized using ETE Toolkit (v3.1.2)[86].

*Phylogenetic analysis of [NiFe] hydrogenase.* Backbone sequences for classifying [NiFe] hydrogenase of Woesearchaeota were retrieved from HydDB[70]. A total of 218 sequences [NiFe] hydrogenase was randomly selected from each subtype and combined with Woesearchaeota sequences. They were aligned with MAFFT-LINSI[74] and trimmed with BMGE (v1.12; default settings)[75]. A tree was generated using IQ-Tree (v1.6.8; settings: -m LG+G -bb 1000)[77].

*Phylogenetic analysis of the rbcL gene.* Backbone sequences of the ribulose-1,5-bisphosphate carboxylase large subunit for classifying Woesearchaeota homologues were retrieved from a previous study[91] and combined with sequences found in Woesearchaeota. Combined sequences were aligned with MAFFT-LINSI[74] and trimmed with BMGE (v1.12; default settings)[75]. A tree was generated using IQ-Tree (v1.6.8; settings: -m LG+G -bb 1000)[77].

*Phylogenetic analysis of 16S rRNA gene.* The 16S rRNA gene sequences of Woesearchaeota and Pacearchaeota were first retrieved from the SILVA SSU 132 database[92], and then combined with sequences from a recent study[18]. The 16S rRNA gene sequences from Woesearchaeota and Pacearchaeota MAGs were identified using barrnap v0.9[93] (https://github.com/tseemann/barrnap). These sequences were then dereplicated using uclust (v1.2.22q) at a threshold of 97%. Sequences that were at least 800 bp–long were retained. A set of 891 16S rRNA gene sequences (35 from Woesearchaeota MAGs, 28 from Pacearchaeota MAGs, and three from unclassified MAGs, 823 from SILVA SSU 132 database or Liu et al.[18]) and two sequences from Nanoarchaeota, used as an outgroup, were included in the phylogenetic analysis. These sequences were aligned using the SINA Alignment Service[94] (https://www.arb-silva.de/aligner/). Full alignments were trimmed with trimAl[95] (v1.4.rev15; settings: -gappyout). A maximum-likelihood tree was then inferred in IQ-Tree (v1.6.8)[77] using TIM+F+G model and node support values were calculated by 1000 ultrafast bootstrap[88]. Woesearchaeota subgroup designations were made based on a previous literature[18].

*Phylogenetic analysis of ribosomal proteins.* The predicted proteins sequences for each genome with adequate quality (≥50% complete and ≤10% contaminated) were searched with hmmsearch[65] (v3.1b2; settings: -E 1e-5) against HMM models provided in GraftM (v0.12.2)[96] representing 15 ribosomal proteins (L2, L3, L5, L6, L10, L11, L14b, L16, S2, S5, S7, S10, S12, S15P, and S19). All proteins with a match to a single copy marker model were aligned using MAFFT-LINSI[74] and automatically trimmed with trimAL (v1.4.rev15; settings: -gappyout)[95]. A maximum-likelihood

tree was inferred using IQ-Tree (v2.0.7; settings: -m LG+C60+F+R -bb 1000 -alrt 1000) based on the concatenation of trimmed 15 ribosomal proteins[79,88].

*Phylogenetic analysis of other individual proteins.* Phylogenetic analysis of other individual protein trees was conducted as described for searching closet relatives of [FeFe] hydrogenase catalytic subunit for Woesearchaeota sequences in the nr database[53]. Detailed models used for each protein could be found in Supplementary Figs. 10–22, 24–41.

**Statistical analysis.** Statistical analysis was done in R version 3.6.0[97] (R development Core Team, Vienna, Austria). One-way analysis of variance (ANOVA) and post-hoc test was conducted to compare the relative abundance of Woesearchaeota across different biotopes using "aov" and 'TukeyHSD' function. The t-distributed stochastic neighbor embedding (t-SNE) analysis and permutational multivariate analysis of variance (PERMANOVA) based on the unweighted UniFrac matrix were completed to test whether Woesearchaeota community shifted among different salinity and biotopes using "Rtsne" in Rtsne package (v0.15) and "adnois" function in vegan package (v2.5-7) respectively. Significance of associations of physiochemical parameters with Woesearchaeota community was assessed using Mantel tests with Pearson's correlation and 999 permutations.

**Reporting summary.** Further information on research design is available in the Nature Research Reporting Summary linked to this article.

## Data availability
The Woesearchaeota MAGs have been deposited in eLMSG (an eLibrary of Microbial Systematics and Genomics, https://www.biosino.org/elmsg/index) and are also available from the NCBI under the BioProject identifier PRJNA746083. The accession number for the MAGs are available in Supplementary Data 1. DNA sequencing data are deposited in the NCBI SRA under the BioProject identifier PRJNA746083 and PRJNA680430. Dataset generated in this study (i.e., protein files for the MAGs, alignments, and tree files) are available through [https://doi.org/10.6084/m9.figshare.14459535]. Public database used in this study included SILVA SSU 132 database [https://www.arb-silva.de/documentation/release-132], Greengenes (v13_8) [http://greengenes.microbio.me/greengenes_release/gg_13_8_otus/], CDD (v3.17) [ftp://ftp.ncbi.nih.gov/pub/mmdb/cdd/], Pfam (v32.0) [ftp://ftp.ebi.ac.uk/pub/databases/Pfam/releases/], SMART (7.1) [https://ftp.ebi.ac.uk/pub/software/unix/iprscan/5/5.38-76.0/], TIGRFAM (v15.0) [ftp://ftp.jcvi.org/pub/data/TIGRFAMs], KEGG KOfam (ver.2020-06-07) [https://www.genome.jp/tools/kofamkoala/], Carbohydrate-active enzymes (CAZymes) database downloaded from dbCAN2 in July 2019 [http://bcb.unl.edu/dbCAN2/download/], MEROPS (v12.0) [ftp://ftp.ebi.ac.uk/pub/databases/merops/old_releases/merops120/], Transporter Classification Database downloaded in November 2020 [http://www.tcdb.org/download.php], eggNOG (v5.0) [http://eggnog5.embl.de/download/eggnog_5.0/], arCOG [http://eggnog5.embl.de/download/eggnog_5.0/per_tax_level/2157/], HydDB [https://services.birc.au.dk/hyddb/], NCBI-nr database downloaded in March 2020 [ftp://ftp.ncbi.nlm.nih.gov/blast/db/] and the archaeal and bacterial backbone datasets published in Dombrowski et al.[36] [https://doi.org/10.5281/zenodo.3672835]. Source data are provided with this paper.

## Code availability
Custom scripts for analyzing phylogenetic trees and ecological analysis have been deposited at [https://doi.org/10.6084/m9.figshare.14459535]. Additionally, we used the published custom code ALE helper scripts [https://github.com/Tancata/phylo/tree/master/ALE] to parse ALE outputs.

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

## Acknowledgements

This work was financially supported by the National Natural Science Foundation of China (grant nos. 91851105, 31970105, 92051102, 91951102, 31622002, 31700430, 91751112, U20A20109); the Shenzhen Science and Technology Program (grant nos. JCYJ20200109105010363, JCYJ20180305163524811, and JCYJ20190808152403587); the Innovation Team Project of Universities in Guangdong Province (No. 2020KCXTD023); and the GDAS' Project of Science and Technology Development (grant no. 2019GDA-SYL-0102003). We thank R.W. and S.-J.K. for support with sampling and preliminary analysis of Jiulong River estuary sediments; M.C. for support with 16S rRNA gene amplicon datasets; and W. L. for assistance in the submission of MAGs to eLMSG.

## Author contributions

W.C.H., Y.L. and M.L. conceived this study. S.Z., W.X., Z.H.L., F.L., Y.L. and W.C.H. participated in sample collection. Y.L. X.Z., D.Z. and C.J.Z. performed metagenomic binning. W.C.H. and Y.L. performed the phylogenetic analysis. W.C.H. performed the evolutionary and metabolic analysis. W.C.H., Y.L. and M.L. wrote the paper. All authors edited and approved the paper.

## Competing interests

The authors declare no competing interests.
