## [Peer Review File · Nature Communications]

Comparative genomic analysis reveals metabolic flexibility of
WoesearchaeotaREVIEWER COMMENTS

Reviewer #1 (Remarks to the Author):

The paper by Huang et al., provides an in depth analysis of the distribution of Woesearchaeota based on 16S rRNA gene data as well as insights into different subclades via metagenome-assembled genomes. Thereby, the authors provide the first in depth overview of this DPANN phylum as well as the first categorization of Woesearchaeota into different phylogenetic subgroups. Additionally, the authors provide a metagenomic analysis and investigate the genomes for HGT events that likely led to the expansion of one of the subgroups.

The authors clearly put an impressive amount of work in this manuscript and investigated Woesearchaeota using a variety of approaches. Especially since most DPANN studies are focused on the whole superphylum, more detailed descriptions of individual phyla are especially needed. However, I do have some comments on the used approaches as well as the details given in the methods sections especially for the phylogenetic analyses and the selection of OGs for the final protein tree:

1. The phylogenetic analyses needs to be described more clearly and I do have concerns about inferring the subgroups mainly on OGs:

a. Have the authors confirmed whether these OGs contain paralogues by running single protein trees? If paralogues were included in the phylogenies there is the risk that these can affect the topology of the species tree. Therefore, I would suggest to explain this part in the methods a bit more clearly and ideally provide single protein trees as a validation that these OGs are suitable to generate concatenated alignments.

b. When establishing marker proteins not only paralogues but also HGT can affect tree topologies¹ and I was wondering if the authors have checked for that as well? Similar to the point above, investigating single protein trees might be a good way to check for that.

c. Can the different subgroups be confirmed across the different trees that were generated? I.e. the authors generated a ribosomal protein tree, a tree based on OGs and a 16S phylogenetic tree. A comparison of these trees and the consistency of the subgroups across them might be something that could be added to the discussion. Additionally, in Supplementary Figure 12 it might be useful to add the color-coding for the subclades.

d. Are the different subclades different classes or orders or are they different taxonomic levels? One way to assess this relatively quickly would be via GTDB_tk, which gives estimates about the ranks.

e. Especially for the HGT analysis, can the authors explain better how the root was chosen and why no non-Woesearchaeota genomes were added/or shown in the analysis leading to Figure 4?

2. The grouping of the different Woesearchaeota genomes into saline (G, I and J) and non-saline groups (H) seems a bit arbitrary. With the exception of the uncharacterized white clade in Figure 2 every single subcluster, including cluster H, includes genomes from saline environments.

Therefore, I would challenge the statement that distinct subgroups and subgroup preferences exist for saline and non-saline environments (Line 125). First, since genomes from saline environments are intermingled what is the evidence for the statement that different subgroups exist for saline/non-saline environments? Second, in SI Figure 3 the non-saline clade H (grey) seems to have a similar peak profile than the saline group I (light green), which would also suggest that salinity might not be the driver of this pattern? A suggestion for this plot would be to change the lines for saline/non-saline groups to be able to see this better or to do this analysis by grouping all saline vs non-saline genomes independent of their subgroup assignment.

3. For the HGT analysis, could the authors add some protein trees, such as for pfk and ackA, to confirm some key inferences that were also discussed in the text? My main reason for this is that HGTector is using sequence similarities to find HGT events and not a phylogenetic approach and thus is depending much more on the accuracy of the database and can lead to false positives.

Some minor comments:

1. Line 60: In the cited study cells were sorted using FACS and then used for metagenome/genome sequencing and the absence of MAGs with DNA contamination from potential hosts was seen as evidence for lack of physical cell-cell associations. The way the

sentence is written it sounds as if this statement is based on microscopic evidence but really this is more indirect evidence and it might be useful to describe this study more clearly in the introduction.

2. Line 77: To my knowledge the evidence for the involvement of Woese archaeota in methane cycling is limited. In the cited study it was shown that Woese archaeota genomes have genes encoding for the MvhD-HdrABC complex, which can be involved in methane metabolism. However, due to the lack of any other gene in this pathway there is no conclusive evidence for the role of Woese archaeota in this pathway. I would suggest to either write this more clearly or remove the reference to methane metabolism. Additionally, have the authors found some additional evidence across the genomes they investigated?

3. Line 107: In the text it states that Woese archaeota abundance ranges from 0.1-4% but in Figure 1a the range goes from 0.001-0.04 (no unit). Is there a reason these numbers are different?

4. Line 131: Since the archaeal root so far is still debated² and there is no outgroup added in Figure 2, I would not state that subgroup A is deeply rooted within the Woese archaeota. Here, this should be either removed or an outgroup or root analysis provided (see issues with the minimal ancestor deviation method below).

5. Lines 168-169: In Figure 3b the two genomes lack at least 4 TCA genes, therefore it is not clear to me why this is seen as a nearly complete TCA pathway? Is it assumed that all 4 genes are absent due to genome completeness? Please clarify.

6. Line 182: What subunits do the Woese archaeota genomes encode? In Data 3 it only lists pdh(subunit) but it is unclear what part of the complex this is and whether the other genes of the complex are lacking? If that is the case there is only limited evidence for the presence of the pyruvate dehydrogenase and this statement should be changed accordingly.

7. Line 200: To my knowledge FeFe hydrogenases are not found typically in archaea³ and can be often misannotated (i.e. to FeS cluster proteins from personal experience). Have the authors validated these hits in any way?

8. Line 208: Why is the focus on subgroup J, can this be explained in the beginning paragraph?

9. Line 401: Were in the analysis does it show that the ancestor of Woese archaeota had fewer genes than extant Woese archaeota? No non-Woese archaeota genomes were included as an outgroup if I interpret this correctly. Therefore can we exclude that the ancestor had more genes, followed by a massive gene loss event?

10. Line 410: This is indeed an interesting feature of group J, however, the lack of lipid biosynthesis genes, other than the ones to synthesize isoprenoids (Line 229), would suggest that they also need to get their lipids from somewhere. Have the authors looked at the number of lipid transporters or do they think that this still is a feature that makes clade J rely on a host? Additionally, it might be interesting to cross-reference this to the HGT analysis. I.e. can the authors predict from where these genes potentially came from?

11. Line 538: If MFP was used in iqtree, please for all captions from the figures and method section include the model that was chosen in the end.

12. Figure 2:

a. Just by reading the methods I realized this was a rooted tree, please add this information also in the caption and also add how the root was chosen.

b. Related to point a: MAD has the problem that it still is affected by long-branch artefacts (personal experience, were adding fast evolving taxa, such as for example Huberarchaeota, very often puts the root with the Huberarchaeota). Could the authors consider confirming the root with alternative methods (i.e. adding an outgroup or using iqtree v2, which allows to use non-time reversible models to infer rooted trees⁴)

c. Environments should be Environments.

d. In the caption please add a description on how the bootstraps were calculated as well as an explanation for the scale bar.

13. Supplementary Figure 3:

a. As mentioned before, the evidence that Subclade A is the ancestor of all Woese archaeota is not clear to me, therefore determining the root position of this clade in that way is not really justifiable. As mentioned before, I would suggest to include an outgroup or try to root the tree.

b. Please include the model used and the bootstrap method.

c. A minor detail, but could the subclade info be added to the figure for easier cross-referencing?

14. Supplementary Data 1: What does 'Statistic' refer to?

15. Supplementary Data 7:

- a. For better accessibility, please add the gene descriptions to the table or at least add the gene names that are also part of Figure 3b.
- b. In the methods, can the authors describe how the OGs were linked to arcogs?

References

1. Bansal, M. S., Wu, Y.-C., Alm, E. J. & Kellis, M. Improved gene tree error correction in the presence of horizontal gene transfer. *Bioinformatics* 31, 1211–1218 (2015).
2. Williams, T. A. et al. Integrative modeling of gene and genome evolution roots the archaeal tree of life. *PNAS* 114, E4602–E4611 (2017).
3. Schuchmann, K., Chowdhury, N. P. & Müller, V. Complex Multimeric [FeFe] Hydrogenases: Biochemistry, Physiology and New Opportunities for the Hydrogen Economy. *Front. Microbiol.* 9, (2018).
4. Minh, B. Q. et al. IQ-TREE 2: New models and efficient methods for phylogenetic inference in the genomic era. *Mol Biol Evol* doi:10.1093/molbev/msaa015.

Reviewer #2 (Remarks to the Author):

In their manuscript, Huang and co-workers explore the environmental distribution and functional potential of Woesearchaeota by mining 16S rRNA specific sequence data in 2,163 available metabarcoding datasets and by comparing 153 metagenome-assembled genomes (MAGs), from which 49 were assembled by the authors. While the analyses presented include considerably more data than previous studies, the ecological and evolutionary analyses are rather superficial and do not precisely or necessarily support the (also rather vague) conclusions. There is a lot of data but the discussion is not well integrated and many assertions are missing confirmatory evidence or a deeper reasoned argumentation.

My major concerns:

- The novelty of the study is limited. The authors simply include more data, but the approach and structure of the manuscript (including the figures) is very similar to a previous manuscript published by some of the authors where they studied 133 clone libraries/studies and 19 publicly available Woesearchaeota genomes (Liu et al., *Microbiome* 2018). One could argue that the addition of new data has led to better constraining the ecology and function of members of the group. Surprisingly, the conclusions of that manuscript were very different than the conclusions of the present work. In the previous work, the authors concluded that Woesearchaeota dominated in anoxic environments, and even suggested syntrophic interactions with methanogenic archaea as potential lifestyle, while here the mention to anaerobic lifestyles is virtually absent and the ecological role of these archaea seems centered around halophily/adaptations to saline environments.
- Distribution of Woesearchaeota in different environments based on 16S rRNA data. To which extent the number of sample types (environments) is not affecting the analysis? Can this be controlled for? Also, concluding that a large phylogenetic group (phylum) is ubiquitous is simplistic and very limited as ecological conclusion. It would be more interesting to study the precise distribution of the different genome types/groups according to the different environments. Unfortunately, the analysis of 16S rRNA metabarcoding data is unlinked from genome data and it is not exploited to the fullest. 16S rRNA data should inform about the diversity of this phylum. However, nothing of this kind is explored. What is the extent of the Woesearchaeota diversity based on 16S rRNA OTUs/data? Where do the MAGs fit in the Woesearchaeota inferred diversity? Do the defined genome clusters correlate with particular 16S sequence types? Are they dispersed in the 16S rRNA phylogenetic tree? Even if 16S rRNA genes are frequently absent from MAGs, the authors may retrieve 16S rRNA sequence reads from the metagenomes they analyze. At the very least, even if they fail to link 16S data to the MAGs, making some type of diversity comparison

should be possible. There is a 16S rRNA tree in Supplementary Fig. 11 but it seems also unlinked from the data the authors analyze and refer to expressed rRNA genes from another study. All this is very rough and unclear.

- The ecological interpretation is vague and questionable. The link with existing literature on microbial ecology and adaptations to the different environments is poor. In particular, the authors highlight the importance of halophilic adaptation and even discuss about the possibility that most Woesearchaeota have a 'salt-out' strategy whereas only a few would have a 'salt-in' strategy based on the proteome pI (proteins from true halophiles are known to be acidic). First of all, the basis of all these assertions is far from clear. The authors describe environments as 'saline' or 'non-saline'. However, the definition of 'saline' is never explicit and, in their supplementary table, they seem to consider as 'saline', marine environments. Between 3.5% (seawater) and ~35% (NaCl-saturating ponds), there is a lot of room for adaptation to various degrees of halophily. Truly halophilic archaea displaying 'salt-in' strategies grow optimally above 20% NaCl. This has nothing to do with halotolerant microbes growing at lower salinities. Marine environments are not considered particularly challenging and needing specific 'salt-out or in' mechanisms. The discussion about all these aspects is extremely poor, speculative and ignores the vast existing literature in the field. The environments must be classified according to their specific salt concentration and so the organisms and their specific adaptations. From the shown pI plots (Suppl. Fig.3), the presence of extreme halophiles is far from clear.

- The annotation of the different genomes allows the authors to confirm previous findings suggesting that these archaea have lost several functions and may rely on hosts as parasites or symbionts. However, genomes from clade J seem to have a larger repertoire of genes. They claim that differences in the number of genes and genome size imply more diversified organisms with flexible metabolism, different from other Woesearchaeota. However, the MAGs are not necessarily complete (>79%) and varying gene contents may relate to various degrees of completion and/or contamination (tolerated values of contamination are quite high, <10% or <5%). Also, the fact that they retain genes involved in nucleotide metabolism and other housekeeping functions does not imply that they have more flexible metabolism in terms of energy transduction. They suggest they might use starch or secrete CAZymes or peptidases and that group J Woesearchaeota might be particle-associated and not parasitic. Unfortunately, evidence for these hypotheses is missing (FISH experiments of these archaea in particles, for instance) or a more substantiated discussion.

- Gene gain and loss, lateral gene transfer (LGT). The authors suggest that there is considerable gene gain in lineage J as compared to other Woesearchaeota. However, inferences about gene gain and loss with relatively partial MAGs (>79%) and with some fraction of contamination (5%) may imply considerable error. Also, the number of non-annotated genes is unknown and it might be that some genes have evolved fast and are no longer easy to recognize as homologs. Gene gain and loss also depend on the outgroup that is considered. If the outgroup archaea are gene-rich, maybe we are in front of gene loss in all but the J clade. All these elements should be considered in a mature and more toned-down discussion.

- In addition, the authors identify cases of LGT using a tool based on similarity. This can be at most used as initial scan for potential genes affected by LGT. However, this is not evidence for LGT. The authors need to provide phylogenetic trees of the corresponding genes with an appropriate taxon sampling in order to show convincing evidence of LGT. While genes with closer homologs in bacteria are potential good candidates for LGT, the authors say that most transferred genes are of DPANN or Euryarchaeota origin, which strongly suggest shared ancestry and not LGT unless otherwise shown by signal-containing phylogenetic trees.

Minor points:

- Which is the percentage of non-annotated genes?
- There are several typos in the manuscript, please check the text.

Reviewer #3 (Remarks to the Author):

In this study, Huang et al analyze Woesearchaeota 16S rRNA gene sequences from EMP, and 153 MAGs (with 49 newly added by this study) to assess biogeography, phylogeny and genomic traits of this clade, including predicted metabolic potential and evolutionary diversification.

The first portion of the manuscript is quite descriptive in nature providing an overview on the global distribution of Woesearchaeota, with the observations that salinity is an important environmental factor for this clade. Phylogenies reveal the 10 subgroups and metabolic predictions reveal novel predictions for saline-specific subgroup J and, most interestingly, the authors suggest that subgroup J shows extensive gene gains for genes related to metabolism and transport of nucleotides and amino acids, which might have been acquired through HGT by archaeal and/or bacterial partners. These data would suggest some level of metabolic flexibility of Woesearchaeota acquired through this genomic expansion and its evolution toward independent lifestyle.

While much of the manuscript is descriptive, the gene gain analysis provides very interesting new insights into the biology of Woesearchaeota, assuming the underlying data supports the conclusions.

Major comments:

Quality of the 153 Woesearchaeota MAGs should be upfront. As is, detailed completeness estimates are only mentioned line 160 when discussing metabolic potential. "We used 153 bona fide Woesearchaeota genomes (104 publicly available and 49 MAGs generated in the current study) for further analyses (see Methods and Supplementary Data 2)." It should be clear upfront and without having to go through the methods and/or supplemental material what the quality metrics of these genomes are, ideally based on CheckM stats (as the authors use per methods) and MIMAG standards. Based on Supplementary Data 2 and the methods, all are at least medium quality (>50% est complete and <10% est contaminated). How did the authors deal with missing markers for phylogeny? This needs to be explained. In the legend of Fig. 2 it should be clarified what the minimum number of markers used was out of the total of 109 markers.

"subgroups G, I, and J seemed to be saline-specific since they consist of genomes predominantly from saline environments," but not exclusively, so this statement has a caveat. The authors also have to consider biases such as the likely lower complexity of metagenomes from saline environments which might better facilitate the successful generation of MAGs, as compared to metagenomes from non-saline environments, which might also lead to underrepresentation of MAGs from non-saline sites in these subgroups. Along these lines: for the proteome isoelectric point analysis (Fig S3), rather than or better in addition to plotting these data by subgroup, could the authors plot it by "environmental metadata" (MAGs from saline versus non-saline environments)?

"To understand the evolutionary relationship between subgroup J and other Woesearchaeota, we selected 47 Woesearchaeotal genomes with a completeness of over 79% and contamination below 5% for further analysis." For orthologous group gain/loss analysis, the level of completeness of the genomes is important. More details should be provided in Figure 4 and its legend (such as completeness estimates of each genome). While nearly 80% provides a more stringent approach as compared to being all-inclusive, there is a caveat to analyze gain/loss with incomplete data, which needs to be carefully assessed by the authors to ensure validity of the results and proper broader interpretation thereof to ensure the underlying data supports the conclusions.

The language throughput could be improved (for some specific examples see below).

Specific and minor comments:

Abstract and throughout:

- "Woesearchaeotal" is adverb and should be lower-case throughout.

Introduction:

- "Archaea, as one of the primary domains of cellular life" – this. Ignores the two-domain scenario. Consider rewording, esp as two domain literature is cited later on in the introduction (ref 4, 5).

- "made possible by the cultivation and bioinformatics methodologies, methodologies, and the continually generated sequencing data" – I'd say primarily due cultivation-independent approaches and advances in sequencing and bioinformatics, the latter two of which go hand in hand. Suggest rewording

- Suggest reducing the overuse of "big words", such as "major" ("major expansion") and "dramatically"

- "The major expansion of the archaeal tree has dramatically" –should better read: archaeal tree of life or archaeal phylogenetic tree

- Figure 1: A, b panel: why not keep the colors consistent for the biotope between the panels?

- Figure 1: Why "Saltmarshes" uppercase; everything else lower case?

- Figure 1: t-SNE analysis of the similarity of woesearchaeotal community matrix is interesting, but some seeming outliers are not explained. Why could the clustering of some saline samples well within the non-saline samples mean? Did the authors investigate potential errors in the metadata. Where exactly did these outlier samples come from?

- Figure 1: "Global distribution of Woesearchaeota. a Global distribution of Woesearchaeota with at least 0.1% relative abundance, based on 2163 16S rRNA gene amplicon datasets." As there are not 2163 visible datapoints on the map, do many datasets have the same coordinates and are thus overlapping? Please clarify for the reader.

- Figure 2: do all MAGs contain all 109 single-copy orthologs? If not clarify.

- Figure 2 legend: describe subgroups A-J. What about datasets that did not fall within a subgroup?

- Line 130: "the average genome size exceeded 1 Mbp" – "estimated genome size" or "assembly size"? As these are MAGs there is no genome size unless it's a complete genome.

- To make a point out the est genome sizes in relation to their evolutionary history, why not add the est genome size information to the phylogenetic tree (outer track, as heat map for example)? Though the GC contents results are not particularly interesting, a GC track could also be added to the phylogeny.

- Line 152: "Whereas, we observed.." – check grammar

- Figure 3 legend (b panel): please add estimated completeness of YT1_182 and Yap2000.bin4.8 to provide better context on what genes/ pathways might be missing due incompleteness versus truly most likely missing in these genomes.

- Line 205: "Collectively, these observations indicate that limited metabolic potentials in carbohydrate metabolism is common among Woesearchaeota." – check grammar

- Line 267: "Consequently, we next evaluated the putative LGT events in 12 high-quality genomes of subgroup J." please define "high quality" or better use MIMAG standards.

REVIEWER COMMENTS

Reviewer #1 (Remarks to the Author):

The paper by Huang et al., provides an in depth analysis of the distribution of Woesearchaeota based on 16S rRNA gene data as well as insights into different subclades via metagenome-assembled genomes. Thereby, the authors provide the first in depth overview of this DPANN phylum as well as the first categorization of Woesearchaeota into different phylogenetic subgroups. Additionally, the authors provide a metagenomic analysis and investigate the genomes for HGT events that likely led to the expansion of one of the subgroups.

The authors clearly put an impressive amount of work in this manuscript and investigated Woesearchaeota using a variety of approaches. Especially since most DPANN studies are focused on the whole superphylum, more detailed descriptions of individual phyla are especially needed. However, I do have some comments on the used approaches as well as the details given in the methods sections especially for the phylogenetic analyses and the selection of OGs for the final protein tree:

Response: Thank you for this fair and constructive evaluation of our work. The reviewer's comments about phylogenetic analysis are valuable and we did additional works to improve our manuscript.

1. The phylogenetic analyses needs to be described more clearly and I do have concerns about inferring the subgroups mainly on OGs:

a. Have the authors confirmed whether these OGs contain paralogues by running single protein trees? If paralogues were included in the phylogenies there is the risk that these can affect the topology of the species tree. Therefore, I would suggest to explain this part in the methods a bit more clearly and ideally provide single protein trees as a validation that these OGs are suitable to generate concatenated alignments.

b. When establishing marker proteins not only paralogues but also HGT can affect tree topologies¹ and I was wondering if the authors have checked for that as well? Similar to the point above, investigating single protein trees might be a good way to check for that.

Response to a-b: We appreciated these two constructive suggestions regarding the improvement of phylogenetic analysis. We annotated the OGs with TIGRFAM and arCOGs (2157 hmm models from eggnog) HMM models according to their best-hits. We first assessed the HGT effects in 109 OGs using a recently published dataset and ranking scheme¹ and then checked paralogues in the OGs used to build a phylogenetic tree. Individual phylogenetic trees are available at figshare (<https://doi.org/10.6084/m9.figshare.14459535>). Based on the newly inferred phylogenetic tree, we reassigned the subgroups. Following the reviewer #1's suggestions, we manually inspected the orthologues used for phylogenetic analysis and selected OGs suitable for concatenation for phylogenetic analysis. Based on the re-inferred tree, we made the following refinement: former subgroup D is now part of subgroup H and I; former subgroup E is now part of subgroup J; former subgroup F is now subgroup D; subgroup E and F are newly assigned.

c. Can the different subgroups be confirmed across the different trees that were generated? I.e. the authors generated a ribosomal protein tree, a tree based on OGs and a 16S phylogenetic tree. A comparison of these trees and the consistency of the subgroups across them might be something that could be added to the discussion. Additionally, in Supplementary Figure 12 it might be useful to add the color-coding for the subclades.

Response: After improving our phylogenetic analysis, we re-inferred the tree for classification of subgroups and discussed the phylogenetic results at Supplementary Note 2. The subgroup A, B, C, G and J are all monophyletic across the three trees (the ribosomal protein tree, the OGs tree and 16S rRNA gene tree). Subgroup D, E, F, H and I are paraphyletic in the 15 ribosomal proteins tree. Subgroup E, I and H are monophyletic in the 16S rRNA gene tree and the ortholog-based trees, although not all MAGs contained 16S rRNA gene sequences and more such sequences are needed to accurately describe their relationship. The color coding was added for the subclades in Supplementary Fig. 5.

d. Are the different subclades different classes or orders or are they different taxonomic levels? One way to assess this relatively quickly would be via GTDB_tk, which gives estimates about the ranks.

Response: We agreed with the reviewer that taxonomy is important for Woesearchaeota. However, taxonomic classification for Woesearchaeota is not the major scope of our paper. We followed your suggestions and estimated the rank of our subgroups (See Table 1 below). According to the GTDB_tk, the subclade comparison was performed at a level between orders to families. Although in GTDB_tk results, different subgroups are at different taxonomic levels, the subgroups were assigned according to the ultrametric tree and the RED value calculated from Fig. 2.

e. Especially for the HGT analysis, can the authors explain better how the root was chosen and why no non-Woesearchaeota genomes were added/or shown in the analysis leading to Figure 4?

Response: Thanks. Considering the Pacearchaeota is phylogenetically most related to Woesearchaeota^{1,2}, they are used as an outgroup. See Figure 5.

2. The grouping of the different Woesearchaeota genomes into saline (G, I and J) and non-saline groups (H) seems a bit arbitrary. With the exception of the uncharacterized white clade in Figure 2 every single subcluster, including cluster H, includes genomes from saline environments. Therefore, I would challenge the statement that distinct subgroups and subgroup preferences exist for saline and non-saline environments (Line 125). First, since genomes from saline environments are intermingled what is the evidence for the statement that different subgroups exist for saline/non-saline environments? Second, in SI Figure 3 the non-saline clade H (grey) seems to have a similar peak profile than the saline group I (light green), which would also suggest that salinity might not be the driver of this pattern? A suggestion for this plot would be to change the lines for saline/non-saline groups to be able to see this better or to do this analysis by grouping all saline vs non-saline genomes independent of their subgroup assignment.

Response: According to your suggestion, we have revised the relevant text on the pI analysis. We removed statements related to saline-specific subgroups, and assigned genomes into saline and non-saline groups accordingly independent of their subgroup assignment. Please see lines 144-155.

3. For the HGT analysis, could the authors add some protein trees, such as for pfk and ackA, to confirm some key inferences that were also discussed in the text? My main reason for this is that HGTector is using sequence similarities to find HGT events and not a phylogenetic approach and thus is depending much more on the accuracy of the database and can lead to false positives.

Response: We have added more protein trees to confirm the key inferences discussed in the text. Please see lines 250-253, 262-264, 272-273, 311-315, 363-366 and 528-531.

Some minor comments:

1. Line 60: In the cited study cells were sorted using FACS and then used for metagenome/genome sequencing and the absence of MAGs with DNA contamination from potential hosts was seen as evidence for lack of physical cell-cell associations. The way the sentence is written it sounds as if this statement is based on microscopic evidence but really this is more indirect evidence and it might be useful to describe this study more clearly in the introduction.

Response: We have added more description of this study in the introduction. See lines 63-67.

2. Line 77: To my knowledge the evidence for the involvement of Woesearchaeota in methane cycling is limited. In the cited study it was shown that Woesearchaeota genomes have genes encoding for the MvhD-HdrABC complex, which can be involved in methane metabolism. However, due to the lack of any other gene in this pathway there is no conclusive evidence for the role of Woesearchaeota in this pathway. I would suggest to either write this more clearly or remove the reference to methane metabolism. Additionally, have the authors found some additional evidence across the genomes they investigated?

Response: We have rephrased this sentence to make it clearer. See lines 82-85. We did not find additional evidence regarding methane metabolism.

3. Line 107: In the text it states that Woesearchaeota abundance ranges from 0.1-4% but in Figure 1a the range goes from 0.001-0.04 (no unit). Is there a reason these numbers are different?

Response: They are the same and units are unified.

4. Line 131: Since the archaeal root so far is still debated² and there is no outgroup added in Figure 2, I would not state that subgroup A is deeply rooted within the Woesearchaeota. Here, this should be either removed or an outgroup or root analysis provided (see issues with the minimal ancestor deviation method below).

Response: Thanks. Outgroup is provided. It indicated that subgroup A is basal to other Woesearchaeota. Please see Figure 2.

5. Lines 168-169: In Figure 3b the two genomes lack at least 4 TCA genes, therefore it is not clear to me why this is seen as a nearly complete TCA pathway? Is it assumed that all 4 genes are absent due to genome completeness? Please clarify.

Response: Modified. See line 189.

6. Line 182: What subunits do the Woesearchaeota genomes encode? In Data 3 it only lists pdh (subunit) but it is unclear what part of the complex this is and whether the other genes of

the complex are lacking? If that is the case there is only limited evidence for the presence of the pyruvate dehydrogenase and this statement should be changed accordingly.

Response: Some genomes in subgroups A, E and J encoded at least three subunits of the pyruvate dehydrogenase (namely *pdhA*, *pdhB*, *pdhC*), indicating pyruvate dehydrogenase is present in these genomes. The subunits of *pdh* encoded by the Woesearchaeota genomes were listed in the Supplementary Data 5.

7. Line 200: To my knowledge FeFe hydrogenases are not found typically in archaea³ and can be often misannotated (i.e. to FeS cluster proteins from personal experience). Have the authors validated these hits in any way?

Response: Thanks for the suggestion. After carefully checking our genomes, we have found that some MAGs belonging to subgroup B, E, G, H and J did encode [FeFe] hydrogenase and it was verified by phylogenetic analysis and their genetic organization (Fig 4, Supplementary Fig 7). More discussion and results are added at lines 225-252 and line 466-481.

8. Line 208: Why is the focus on subgroup J, can this be explained in the beginning paragraph?

Response: Thanks. We have extended the explanation of our focus on subgroup J. Please see lines 322-326.

9. Line 401: Were in the analysis does it show that the ancestor of Woesearchaeota had fewer genes than extant Woesearchaeota? No non-Woesearchaeota genomes were included as an outgroup if I interpret this correctly. Therefore can we exclude that the ancestor had more genes, followed by a massive gene loss event?

Response: Indeed, we could not rule out this possibility. Therefore, this statement has been removed from the text.

10. Line 410: This is indeed an interesting feature of group J, however, the lack of lipid biosynthesis genes, other than the ones to synthesize isoprenoids (Line 229), would suggest that they also need to get their lipids from somewhere. Have the authors looked at the number of lipid transporters or do they think that this still is a feature that makes clade J rely on a host? Additionally, it might be interesting to cross-reference this to the HGT analysis. I.e. can the authors predict from where these genes potentially came from?

Response: Although complete lipid biosynthesis pathway is absent in the genomes of subgroup J, two most complete genomes (YT1_182 and Yap2000.bin.4) in subgroup J encode *gds*, *carS* and *pssA* gene involved in the archaeol biosynthesis and we queried our protein sequences against TCDB database. They appeared to lack transporters for lipids and indeed may need to get their lipids from somewhere else. However, from their genomic contents, we think subgroup J is more independent compared to other Woesearchaeota subgroups. We inferred phylogenetic trees for individual genes in the MEP pathway and they appeared to have bacterial originations. See line 310-314.

11. Line 538: If MFP was used in iqtree, please for all captions from the figures and method section include the model that was chosen in the end.

Response: This information is added in all captions and method section if MFP was used.

12. Figure 2:

a. Just by reading the methods I realized this was a rooted tree, please add this information also in the caption and also add how the root was chosen.

b. Related to point a: MAD has the problem that it still is affected by long-branch artefacts (personal experience, were adding fast evolving taxa, such as for example Huberarchaeota, very often puts the root with the Huberarchaeota). Could the authors consider confirming the root with alternative methods (i.e., adding an outgroup or using iqtree v2, which allows to use non-time reversible models to infer rooted trees⁴)

c. Environments should be Environments.

d. In the caption please add a description on how the bootstraps were calculated as well as an explanation for the scale bar.

Response: Thanks. We re-inferred the tree for subgroup division and rooted the tree using Paccarchaeota as an outgroup. How bootstraps were calculated and explanation for the scale bar were also added in the captions for Figure 2. Other error collections have also been done.

13. Supplementary Figure 3:

a. As mentioned before, the evidence that Subclade A is the ancestor of all Woesearchaeota is not clear to me, therefore determining the root position of this clade in that way is not really justifiable. As mentioned before, I would suggest to include an outgroup or try to root the tree.

b. Please include the model used and the bootstrap method.

c. A minor detail, but could the subclade info be added to the figure for easier cross-referencing?

Response: Yes. We have updated the analysis and added outgroup sequences to root the tree. Model and subclade info are added.

14. Supplementary Data 1: What does 'Statistic' refer to?

Response: It refers to Mantel test statistic (r), ranging from -1 (negative) to 0 (no effect) to 1 (positive), which measures the strength of the relationship between physiochemical parameters and Woesearchaeota community.

15. Supplementary Data 7:

a. For better accessibility, please add the gene descriptions to the table or at least add the gene names that are also part of Figure 3b.

b. In the methods, can the authors describe how the OGs were linked to arcogs?

Response: Gene descriptions were added to the Supplementary Data 7 and the approach for linking the OGs to arCOGs is also described in the methods.

References

1. Bansal, M. S., Wu, Y.-C., Alm, E. J. & Kellis, M. Improved gene tree error correction in the presence of horizontal gene transfer. *Bioinformatics* 31, 1211–1218 (2015).
2. Williams, T. A. et al. Integrative modeling of gene and genome evolution roots the archaeal tree of life. *PNAS* 114, E4602–E4611 (2017).

3. Schuchmann, K., Chowdhury, N. P. & Müller, V. Complex Multimeric [FeFe] Hydrogenases: Biochemistry, Physiology and New Opportunities for the Hydrogen Economy. *Front. Microbiol.* 9, (2018).
4. Minh, B. Q. et al. IQ-TREE 2: New models and efficient methods for phylogenetic inference in the genomic era. *Mol Biol Evol* doi:10.1093/molbev/msaa015.

References:

1. Dombrowski, N. et al. Undinarchaeota illuminate DPANN phylogeny and the impact of gene transfer on archaeal evolution. *Nature Communications* 11, 3939 (2020).
2. Castelle, C. J. et al. Biosynthetic capacity, metabolic variety and unusual biology in the CPR and DPANN radiations. *Nature Reviews Microbiology* 16, 629–645 (2018).

Reviewer #2 (Remarks to the Author):

In their manuscript, Huang and co-workers explore the environmental distribution and functional potential of Woesearchaeota by mining 16S rRNA specific sequence data in 2,163 available metabarcoding datasets and by comparing 153 metagenome-assembled genomes (MAGs), from which 49 were assembled by the authors. While the analyses presented include considerably more data than previous studies, the ecological and evolutionary analyses are rather superficial and do not precisely or necessarily support the (also rather vague) conclusions. There is a lot of data but the discussion is not well integrated and many assertions are missing confirmatory evidence or a deeper reasoned argumentation.

Response: We appreciate the reviewer's comments. We have revised the manuscript as advised by all reviewers. We believed that the manuscript has improved and related ecological and evolutionary analyses are also improved accordingly.

My major concerns:

- The novelty of the study is limited. The authors simply include more data, but the approach and structure of the manuscript (including the figures) is very similar to a previous manuscript published by some of the authors where they studied 133 clone libraries/studies and 19 publicly available Woesearchaeota genomes (Liu et al., *Microbiome* 2018). One could argue that the addition of new data has led to better constraining the ecology and function of members of the group. Surprisingly, the conclusions of that manuscript were very different than the conclusions of the present work. In the previous work, the authors concluded that

Woesearchaeota dominated in anoxic environments, and even suggested syntrophic interactions with methanogenic archaea as potential lifestyle, while here the mention to anaerobic lifestyles is virtually absent and the ecological role of these archaea seems centered around halophily/adaptations to saline environments.

Response: As summarized by Reviewer #2, this manuscript is mining more comprehensive, metabarcoding and broadly not only covered 16S rRNA gene datasets from Earth Microbiome Project but also much more metagenome-assembled genomes than Liu et al., Microbiome 2018. We have documented the methods there and here, generated novel discovery and analytic results to share with broad readerships. We consider our findings here as a complement and update to Liu et al., Microbiome 2018. Though the number of metabarcoding libraries with oxygen parameters in the selected EMP dataset is limited (n=37), mantel test showed oxygen has significant impacts on Woesearchaeota community ($P < 0.05$). More importantly, we have updated the results and revised the text related to a predicted anaerobic lifestyles of subgroup J. The presence of [FeFe] hydrogenase may indicate that they live an anerobic lifestyle, which is a new finding in the current study. Please see lines 225-281.

- Distribution of Woesearchaeota in different environments based on 16S rRNA data. To which extent the number of sample types (environments) is not affecting the analysis? Can this be controlled for? Also, concluding that a large phylogenetic group (phylum) is ubiquitous is simplistic and very limited as ecological conclusion. It would be more interesting to study the precise distribution of the different genome types/groups according to the different environments. Unfortunately, the analysis of 16S rRNA metabarcoding data is unlinked from genome data and it is not exploited to the fullest. 16S rRNA data should inform about the diversity of this phylum. However, nothing of this kind is explored. What is the extent of the Woesearchaeota diversity based on 16S rRNA OTUs/data? Where do the MAGs fit in the Woesearchaeota inferred diversity? Do the defined genome clusters correlate with particular 16S sequence types? Are they dispersed in the 16S rRNA phylogenetic tree? Even if 16S rRNA genes are frequently absent from MAGs, the authors may retrieve 16S rRNA sequence reads from the metagenomes they analyze. At the very least, even if they fail to link 16S data to the MAGs, making some type of diversity comparison should be possible. There is a 16S rRNA tree in Supplementary Fig. 11 but it seems also unlinked from the data the authors analyze and refer to expressed rRNA genes from another study. All this is very rough and unclear.

Response: Thanks for the comments. The distribution of Woesearchaeota in different environments is an ecological investigation and the number of sample types was not controlled. We agreed that the study of precise distribution of the genome types/groups is more interesting and in the revised paper, we have linked 16S rRNA gene sequence cluster, Woese-3, Woese-4, Woese-14b, Woese-14a, Woese-24 and Woese-21a to subgroups A, C, E, H, I and J, respectively. In addition, subgroup G has sequence representatives in a monophyletic clade in the 16S rRNA gene tree including Woese-8, Woese-10, Woese-9, Woese-6, Woese-18 and Woese-20. The sequence clusters were used to probe the approximate distribution of subgroups in the biotopes investigated. The analysis showed subgroup G, I and J were present in all biotopes investigated and thus may have high ecological adaptability. See also lines 171-182.

- The ecological interpretation is vague and questionable. The link with existing literature on microbial ecology and adaptations to the different environments is poor. In particular, the authors highlight the importance of halophilic adaptation and even discuss about the possibility that most Woesearchaeota have a 'salt-out' strategy whereas only a few would

have a 'salt-in' strategy based on the proteome pI (proteins from true halophiles are known to be acidic). First of all, the basis of all these assertions is far from clear. The authors describe environments as 'saline' or 'non-saline'. However, the definition of 'saline' is never explicit and, in their supplementary table, they seem to consider as 'saline', marine environments. Between 3.5% (seawater) and ~35% (NaCl-saturating ponds), there is a lot of room for adaptation to various degrees of halophily. Truly halophilic archaea displaying 'salt-in' strategies grow optimally above 20% NaCl. This has nothing to do with halotolerant microbes growing at lower salinities. Marine environments are not considered particularly challenging and needing specific 'salt-out or in' mechanisms. The discussion about all these aspects is extremely poor, speculative and ignores the vast existing literature in the field. The environments must be classified according to their specific salt concentration and so the organisms and their specific adaptations. From the shown pI plots (Suppl. Fig.3), the presence of extreme halophiles is far from clear.

Response: Thanks for the comments. First of all, we did not claim any Woesearchaeota as halophiles in the current study. We simply grouped each genome according to their sampling environment, i.e., "groundwater" and "freshwater" samples are considered as "non-saline", while "saline water", "mangrove", "seagrass bed", "marine water/sediment", "soda lake water/sediment", "estuary sediment", and "hydrothermal vent" are considered as "saline". This definition is quite loosed for simplifying the analysis. Genomes sampled from hot spring, waste water and soil were excluded from the pI analysis. We are aware of vast existing literature in the field of genome content variation towards saline adaption. Previous knowledge is mainly based on cultured strains until Cabello-Yeves and Rodriguez-Valera (Microbiome 2019 7:117)¹ gave a comprehensive analysis of large-scaled metaproteomes. It is stated that "the exact physiological explanation for such variations in the pIs and electrostatic surface potentials is not known yet". Since our ecological analysis revealed that salinity levels were the most significantly correlated with the Woesearchaeota community composition, we naturally dived into the genome content to see if there are variations towards the salinity changes, in our case, saline or non-saline habitats. We have revised relevant texts (see Lines 144-155) and figure (see Supplementary Fig. 3).

- The annotation of the different genomes allows the authors to confirm previous findings suggesting that these archaea have lost several functions and may rely on hosts as parasites or symbionts. However, genomes from clade J seem to have a larger repertoire of genes. They claim that differences in the number of genes and genome size imply more diversified organisms with flexible metabolism, different from other Woesearchaeota. However, the MAGs are not necessarily complete (>79%) and varying gene contents may relate to various degrees of completion and/or contamination (tolerated values of contamination are quite high, <10% or <5%). Also, the fact that they retain genes involved in nucleotide metabolism and other housekeeping functions does not imply that they have more flexible metabolism in terms of energy transduction. They suggest they might use starch or secrete CAZymes or peptidases and that group J Woesearchaeota might be particle-associated and not parasitic.

Unfortunately, evidence for these hypotheses is missing (FISH experiments of these archaea in particles, for instance) or a more substantiated discussion.

Response: Following the suggestions of other reviewers, we re-inferred the phylogenetic trees for subgroup assignments and subgroup J included former subgroup E (See response to reviewer 1). The metabolic analysis indicated subgroup J also had features indicating more flexible energy transduction like the complete glycolytic pathway, [FeFe] hydrogenase, Rnf complex and V-type ATPase. The current completeness is evaluated by CheckM, which estimated the first circular Woesearchaeota genome - GW2011_AR20 to be 79.17% complete,

indicating the marker-set used by CheckM might not provide the *bona fide* completeness for some lineages. Therefore, we believed the completeness value is a relatively fair reference but not an absolute standard to evaluate the MAG quality. While the tolerated values of contamination (< 10% or < 5%) seemed high, it is generally accepted in comparative genomic analysis of MAGs²⁻⁴. We found additional evidence compensating the glycolytic pathway to support that they might be able to produce energy independently. The evidences for these hypotheses are, of no doubt, necessary, and are part of our future research plans. To improve this point, we have added more discussion accordingly (Line 486-487).

- Gene gain and loss, lateral gene transfer (LGT). The authors suggest that there is considerable gene gain in lineage J as compared to other Woesearchaeota. However, inferences about gene gain and loss with relatively partial MAGs (>79%) and with some fraction of contamination (5%) may imply considerable error. Also, the number of non-annotated genes is unknown and it might be that some genes have evolved fast and are no longer easy to recognize as homologs. Gene gain and loss also depend on the outgroup that is considered. If the outgroup archaea are gene-rich, maybe we are in front of gene loss in all but the J clade. All these elements should be considered in a mature and more toned-down discussion.

Response: Thanks again for the comments. We believed these MAGs are relatively complete by referring to the GW2011_AR20. Furthermore, the inclusion of MAGs (> 79% complete and < 5% contaminated) might be also acceptable for other studies, such as a lower quality metric (> 45% and < 10%) was used in a recent paper published on *Nature Communications*⁵. In addition, we used an approach which probabilistically accounts for the missing fraction of the genomes in the revised paper. The outgroup, Pacearchaeota, is also selected because they are most phylogenetically related to Woesearchaeota^{5,6}.

- In addition, the authors identify cases of LGT using a tool based on similarity. This can be at most used as initial scan for potential genes affected by LGT. However, this is not evidence for LGT. The authors need to provide phylogenetic trees of the corresponding genes with an appropriate taxon sampling in order to show convincing evidence of LGT. While genes with closer homologs in bacteria are potential good candidates for LGT, the authors say that most transferred genes are of DPANN or Euryarchaeota origin, which strongly suggest shared ancestry and not LGT unless otherwise shown by signal-containing phylogenetic trees.

Response: We have updated the LGT analysis by including more phylogenetic analysis. The candidate sequences for analysis were selected by comprehensive search in the NR database and we believed the analysis is improved. (Lines 719-731, lines 747-751)

Minor points:

- Which is the percentage of non-annotated genes?

Response: Shown in Fig. 5b.

- There are several typos in the manuscript, please check the text.

Response: Corrected.

References:

1. Cabello-Yeves, P. J. & Rodriguez-Valera, F. Marine-freshwater prokaryotic transitions require extensive changes in the predicted proteome. *Microbiome* 7, 117 (2019).

2. Tully, B. J. Metabolic diversity within the globally abundant Marine Group II Euryarchaea offers insight into ecological patterns. *Nature Communications* 10, 271 (2019).
3. Sheridan, P. O. et al. Gene duplication drives genome expansion in a major lineage of Thaumarchaeota. *Nature Communications* 11, 5494 (2020).
4. Rinke, C. et al. A phylogenomic and ecological analysis of the globally abundant Marine Group II archaea (Ca . Poseidoniales ord. nov.). *ISME J* 13, 663–675 (2019).
5. Castelle, C. J. et al. Biosynthetic capacity, metabolic variety and unusual biology in the CPR and DPANN radiations. *Nature Reviews Microbiology* 16, 629–645 (2018).
6. Dombrowski, N. et al. Undinarchaeota illuminate DPANN phylogeny and the impact of gene transfer on archaeal evolution. *Nature Communications* 11, 3939 (2020).

Reviewer #3 (Remarks to the Author):

In this study, Huang et al analyze Woesearchaeota 16S rRNA gene sequences from EMP, and 153 MAGs (with 49 newly added by this study) to assess biogeography, phylogeny and genomic traits of this clade, including predicted metabolic potential and evolutionary diversification.

The first portion of the manuscript is quite descriptive in nature providing an overview on the global distribution of Woesearchaeota, with the observations that salinity is an important environmental factor for this clade. Phylogenies reveal the 10 subgroups and metabolic predictions reveal novel predictions for saline-specific subgroup J and, most interestingly, the authors suggest that subgroup J shows extensive gene gains for genes related to metabolism and transport of nucleotides and amino acids, which might have been acquired through HGT by archaeal and/or bacterial partners. These data would suggest some level of metabolic flexibility of Woesearchaeota acquired through this genomic expansion and its evolution toward independent lifestyle.

While much of the manuscript is descriptive, the gene gain analysis provides very interesting new insights into the biology of Woesearchaeota, assuming the underlying data supports the conclusions.

Major comments:

Quality of the 153 Woese archaeota MAGs should be upfront. As is, detailed completeness estimates are only mentioned line 160 when discussing metabolic potential. “We used 153 bona fide Woese archaeota genomes (104 publicly available and 49 MAGs generated in the current study) for further analyses (see Methods and Supplementary Data 2).” It should be clear upfront and without having to go through the methods and/or supplemental material what the quality metrics of these genomes are, ideally based on CheckM stats (as the authors use per methods) and MIMAG standards. Based on Supplementary Data 2 and the methods, all are at least medium quality (>50% est complete and <10% est contaminated). How did the authors deal with missing markers for phylogeny? This needs to be explained. In the legend of Fig. 2 it should be clarified what the minimum of markers used was out of the total of 109 markers.

Response: Thank you. The quality metrics about MAGs used in this study was added to the front (See line 133-143) and in the legend of Fig. 2, more description was added to the minimum of markers used.

“subgroups G, I, and J seemed to be saline-specific since they consist of genomes predominantly from saline environments,” but not exclusively, so this statement has a caveat. The authors also have to consider biases such as the likely lower complexity of metagenomes from saline environments which might better facilitate the successful generation of MAGs, as compared to metagenomes from non-saline environments, which might also lead to underrepresentation of MAGs from non-saline sites in these subgroups. Along these lines: for the proteome isoelectric point analysis (Fig S3), rather than or better in addition to plotting these data by subgroup, could the authors plot it by “environmental metadata” (MAGs from saline versus non-saline environments)?

Response: We agreed and modified this part. Please see line 143-154.

“To understand the evolutionary relationship between subgroup J and other Woese archaeota, we selected 47 Woese archaeotal genomes with a completeness of over 79% and contamination below 5% for further analysis.” For orthologous group gain/loss analysis, the level of completeness of the genomes is important. More details should be provided in Figure 4 and its legend (such as completeness estimates of each genome). While nearly 80% provides a more stringent approach as compared to being all-inclusive, there is a caveat to analyze gain/loss with incomplete data, which needs to be carefully assessed by the authors to ensure validity of the results and proper broader interpretation thereof to ensure the underlying data supports the conclusions.

Response: We agreed that completeness is important for the gain/loss analysis. However, the completeness is assessed by the presence specific markers, which may not represent the *bona fide* completeness of the genomes, and we chose this criterion according to the first circular Woese archaeota MAGs reconstructed in Castelle et al. 2015¹. Second, in the revised paper, we used Amalgamated Likelihood Estimation (ALE) package, a gene-tree-aware approach, in the gain/loss analysis this time. This software incorporated a probabilistic approach to account for the missing fraction of the genomes and was also used to perform gain/loss analysis in incomplete data elsewhere².

The language throughout could be improved (for some specific examples see below).

Specific and minor comments:

Abstract and throughout:

- “Woese archaeotal” is adverb and should be lower-case throughout.

Response: This typo has been corrected.

Introduction:

- “Archaea, as one of the primary domains of cellular life” – this. Ignores the two-domain scenario. Consider rewording, esp as two domain literature is cited later on in the introduction (ref 4, 5).

Response: Modified. See line 46-47.

- “made possible by the cultivation and bioinformatics methodologies, methodologies, and the continually generated sequencing data” – I’d say primarily due cultivation-independent approaches and advances in sequencing and bioinformatics, the latter two of which go hand in hand. Suggest rewording

Response: Reworded as suggested. See line 49-50.

- Suggest reducing the overuse of “big words”, such as “major” (“major expansion”) and “dramatically”

Response: Reduced.

- “The major expansion of the archaeal tree has dramatically” –should better read: archaeal tree of life or archaeal phylogenetic tree

Response: These sentences have been reworded following the reviewer’s suggestions. See line 50.

- Figure 1: A, b panel: why not keep the colors consistent for the biotope between the panels?

Response: The colors were kept consistent.

- Figure 1: Why “Saltmarshes” uppercase; everything else lower case?

Response: We have kept it consistent with others.

- Figure 1: t-SNE analysis of the similarity of woese archaeotal community matrix is interesting, but some seeming outliers are not explained. Why could the clustering of some saline samples well within the non-saline samples mean? Did the authors investigate potential errors in the metadata. Where exactly did these outlier samples come from?

Response: The three samples that clustered well within the non-saline samples, were collected from hydrothermal environments of Brazelton Lost City, located in the Mid-Atlantic Ridge. Therefore, we suspected that these samples may represent ancient Woese archaeota community which then migrated to non-saline environments.

- Figure 1: “Global distribution of Woese archaeota. a Global distribution of Woese archaeota with at least 0.1% relative abundance, based on 2163 16S rRNA gene amplicon datasets.” As

there are not 2163 visible datapoints on the map, do many datasets have the same coordinates and are thus overlapping? Please clarify for the reader.

Response: Yes, some datasets have the same coordinates and are therefore overlapping. The information was added in the Figure legends.

- Figure 2: do all MAGs contain all 109 single-copy orthologs? If not clarify.

Response: We have updated the phylogeny and added the marker info to the Figure legend 2.

- Figure 2 legend: describe subgroups A-J. What about datasets that did not fall within a subgroup?

Response: Added in the legends.

- Line 130: “the average genome size exceeded 1 Mbp” – “estimated genome size” or “assembly size”? As these are MAGs there is no genome size unless it’s a complete genome.

Response: Corrected.

- To make a point out the est genome sizes in relation to their evolutionary history, why not add the est genome size information to the phylogenetic tree (outer track, as heat map for example)? Though the GC contents results are not particularly interesting, a GC track could also be added to the phylogeny.

Response: Added.

- Line 152: “Whereas, we observed.” – check grammar

Response: Checked.

- Figure 3 legend (b panel): please add estimated completeness of YT1_182 and Yap2000.bin4.8 to provide better context on what genes/ pathways might be missing due to incompleteness versus truly most likely missing in these genomes.

Response: The estimated completeness was added to the figure legends.

- Line 205: “Collectively, these observations indicate that limited metabolic potentials in carbohydrate metabolism is common among Woesearchaeota.” – check grammar

Response: Checked.

- Line 267: “Consequently, we next evaluated the putative LGT events in 12 high-quality genomes of subgroup J.” please define “high quality” or better use MIMAG standards.

Response: This term was not used in the revised paper.

References:

1. Castelle, C. J. et al. Genomic expansion of domain archaea highlights roles for organisms from new phyla in anaerobic carbon cycling. *Curr. Biol.* 25, 690–701 (2015).
2. Sheridan, P. O. et al. Gene duplication drives genome expansion in a major lineage of Thaumarchaeota. *Nature Communications* 11, 5494 (2020).

Table 1: Taxonomic status of Woesearchaeota estimated by GTDB-tk using the R95 release

MAG id	Subgroup	GTDDB order	GTDDB family	GTDDB genus	GTDDB species
GW2011_AR18	A	SCGC-AAA011-G17	GW2011-AR18	GW2011-AR18	GW2011-AR18 sp000806155
GW2011_AR20	A	SCGC-AAA011-G17	GW2011-AR20	GW2011-AR20	GW2011-AR20 sp000830315
UBA489	A	SCGC-AAA011-G17	UBA489	UBA489	UBA489 sp002505585
UBA544	A	SCGC-AAA011-G17	UBA489	UBA489	UBA489 sp002505585
GW2011_AR17	A	SCGC-AAA011-G17	GW2011-AR17	GW2011-AR17	GW2011-AR17 sp10136u
J104	A	SCGC-AAA011-G17	GW2011-AR17	J104	J104 sp003694495
MP5_9_10	B	Woesearchaeales	B72-G16		
FT2_175	B	Woesearchaeales	B72-G16		
YT1_092	B	Woesearchaeales			
Yap200.bin8.246	B	Woesearchaeales	B72-G16		
FT2_141	B	Woesearchaeales	B72-G16		
MP5_1_30	B	Woesearchaeales	B72-G16		
JLRS3_6_219	B	Woesearchaeales	B72-G16		
FT6_98	B	Woesearchaeales	B72-G16		
CG10_big_fil_rev_8_21_14_0_10_36_11	C	Woesearchaeales	GW2011-AR9	PCYB01	PCYB01 sp002762845
CG10_big_fil_rev_8_21_14_0_10_45_16	C	Woesearchaeales	GW2011-AR9	UBA11998	UBA11998 sp002762785
UBA11998	C	Woesearchaeales	GW2011-AR9	UBA11998	UBA11998 sp11998u
ARS1334	C	Woesearchaeales	GW2011-AR9	UBA11998	UBA11998 sp002686315
CG10_big_fil_rev_8_21_14_0_10_32_24	C	Woesearchaeales	GW2011-AR9	1-14-0-10-32-24	1-14-0-10-32-24 sp002763025
UBA10204	C	Woesearchaeales	GW2011-AR9	UBA10204	UBA10204 sp10204u
UBA10194	C	Woesearchaeales	GW2011-AR9	GW2011-AR9	GW2011-AR9 sp10194u
CG_4_10_14_0_2_um_filter_33_13	C	Woesearchaeales	GW2011-AR9	GCA-002792115	GCA-002792115 sp002792115
ARS1203	C	Woesearchaeales	GW2011-AR9	GCA-2688265	GCA-2688265 sp002688265
1420	C	Woesearchaeales	GW2011-AR9	NZBF01	NZBF01 sp002687775
FT1_228	C	Woesearchaeales			
YT3_319	D	Woesearchaeales	UBA11576		
JLRS2_1_154	D	Woesearchaeales	UBA11576		
FT2_479	D	Woesearchaeales	UBA11576		
FT1_219	D	Woesearchaeales	UBA11576	UBA11576	
UBA11576	D	Woesearchaeales	UBA11576	UBA11576	UBA11576 sp11576u
J101	D	Woesearchaeales	UBA11576	J101	J101 sp003695435
CG10_big_fil_rev_8_21_14_0_10_44_13	D	Woesearchaeales	UBA12501	1-14-0-10-44-13	1-14-0-10-44-13 sp002762985
UBA10192	D	Woesearchaeales	UBA12501	UBA12501	UBA12501 sp10192u
YT2_062	D	Woesearchaeales			
MP5_7_64	D	Woesearchaeales			
UBA10107	D	Woesearchaeales	UBA10107	UBA10107	UBA10107 sp10107u
CG11_big_fil_rev_8_21_14_0_20_43_8	D	Woesearchaeales	CG08-08-20-14	CG08-08-20-14	CG08-08-20-14 sp002762705
CG08_lamd_8_20_14_0_20_43_7	D	Woesearchaeales	CG08-08-20-14	CG08-08-20-14	CG08-08-20-14 sp002762705
CG11_big_fil_rev_8_21_14_0_20_57_5	E	Woesearchaeales	CG1-02-57-44	CG1-02-57-44	CG1-02-57-44 sp001871415
CG_4_10_14_0_2_um_filter_57_5	E	Woesearchaeales	CG1-02-57-44	CG1-02-57-44	CG1-02-57-44 sp001871415
UBA94	E	Woesearchaeales	CG1-02-57-44	CG1-02-57-44	CG1-02-57-44 sp001871415
CG1_02_57_44	E	Woesearchaeales	CG1-02-57-44	CG1-02-57-44	CG1-02-57-44 sp001871415
GW2011_AR4	E	Woesearchaeales	UBA9989	UBA9989	UBA9989 sp9989u
UBA9989	E	Woesearchaeales	UBA9989	UBA9989	UBA9989 sp9989u
TSed10_208R1	E	Woesearchaeales	J072		
CG10_big_fil_rev_8_21_14_0_10_34_8	E	Woesearchaeales	J091		
J091	E	Woesearchaeales	J091	J091	J091 sp003695045
CSBr16_68R1	F	Woesearchaeales	B72-G16		
CSBr16_144	F	Woesearchaeales	UBA10107		
CSBr16_28	F	Woesearchaeales	UBA10107		
UBA153	G	Woesearchaeales	UBA525	UBA153	UBA153 sp002503705
J116	G	Woesearchaeales	UBA525	UBA153	UBA153 sp003694385
UBA10207	G	Woesearchaeales	UBA525	UBA525	UBA525 sp10207u
UBA525	G	Woesearchaeales	UBA525	UBA525	UBA525 sp002498125
J110	G	Woesearchaeales	J110	J110	J110 sp003694805
CSSed10_238R1	G	Woesearchaeales	21-14-0-10-32-9	PWWA01	PWWA01 sp003560545
CSSed165cm_557	G	Woesearchaeales	21-14-0-10-32-9	PWWA01	PWWA01 sp007131205
CSSed10_415	G	Woesearchaeales	21-14-0-10-32-9	PWWA01	PWWA01 sp007131205
CSSed11_337R1	G	Woesearchaeales	21-14-0-10-32-9	PWWA01	PWWA01 sp003562145
CSSed10_383	G	Woesearchaeales	21-14-0-10-32-9	21-14-0-10-32-9	
CG10_big_fil_rev_8_21_14_0_10_32_9	G	Woesearchaeales	21-14-0-10-32-9	21-14-0-10-32-9	21-14-0-10-32-9 sp002762915
CSBr16_111R1	G	Woesearchaeales	21-14-0-10-32-9	SKIS01	SKIS01 sp007117755
CSBr16_71	G	Woesearchaeales	21-14-0-10-32-9	SKIS01	SKIS01 sp007116295
CSBr16_182	G	Woesearchaeales	21-14-0-10-32-9	CSBR16-182	CSBR16-182 sp007117145
Yap75.bin7.84	G	Woesearchaeales	21-14-0-10-32-9		
CSBr16_114	G	Woesearchaeales	SKGA01	SKGA01	SKGA01 sp007117735
ARS73	G	Woesearchaeales	GCA-2685855	GCA-2685855	GCA-2685855 sp002685855
ARS1441	G	Woesearchaeales	GCA-2686295	GCA-2686295	GCA-2686295 sp002686295
SM23-78	G	Woesearchaeales	SM23-78	SM23-78	SM23-78 sp001595785
Yap2000.bin8.138	G	Woesearchaeales	SM23-78	SM23-78	SM23-78 sp001595785
YT2_088	G	Woesearchaeales	SM23-78	SM23-78	
UBA12459	G	Woesearchaeales	UBA9642	UBA9642	UBA9642 sp12459u
SURF_58	G	Woesearchaeales	UBA9642	SURF-58	SURF-58 sp003599145
CSBr16_197	G	Woesearchaeales	SKIA01	SKIA01	SKIA01 sp007117065
CSSed162cmB_258	G	Woesearchaeales	SKIA01	SKIA01	SKIA01 sp007128245

CSBr16_223	G	Woesearchaeales	SKIA01	SKIA01	SKIA01	sp007128245
CSBr16_4	G	Woesearchaeales	SKIA01	SKIA01	SKIA01	sp007116645
T3Sed10_350R1	G	Woesearchaeales	PXDW01	PXDW01	PXDW01	sp003564925
BM511	G	Woesearchaeales	BM511	BM511	BM511	sp002867475
GW2011_AR15	G	Woesearchaeales	GW2011-AR15	GW2011-AR15	GW2011-AR15	sp000830295
CG10_big_fil_rev_8_21_14_0_10_47_5	H	Woesearchaeales	CG1-02-47-18	CG1-02-47-18	CG1-02-47-18	sp002763335
CG1_02_47_18	H	Woesearchaeales	CG1-02-47-18	CG1-02-47-18	CG1-02-47-18	sp002763335
CG08_land_8_20_14_0_20_47_9	H	Woesearchaeales	CG1-02-47-18	CG1-02-47-18	CG1-02-47-18	sp002763335
UBA142	H	Woesearchaeales	UBA10216	UBA492	UBA492	sp002688315
ARS1199	H	Woesearchaeales	UBA10216	UBA492	UBA492	sp002688315
UBA492	H	Woesearchaeales	UBA10216	UBA492	UBA492	sp002688315
UBA10216	H	Woesearchaeales	UBA10216	UBA10216	UBA10216	sp10216u
CG10_big_fil_rev_8_21_14_0_10_45_5	H	Woesearchaeales	0-14-0-80-44-23	0-14-0-80-44-23	0-14-0-80-44-23	sp002779235
CG07_land_8_20_14_0_80_44_23	H	Woesearchaeales	0-14-0-80-44-23	0-14-0-80-44-23	0-14-0-80-44-23	sp002779235
FT1_346	H	Woesearchaeales	CG08-08-20-14			
CG_4_10_14_0_8_um_filter_47_5	H	Woesearchaeales	CG08-08-20-14			
B29_G15	H	Woesearchaeales	B29-G15	B29-G15	B29-G15	sp003650585
Yap30.bin1.57	H	Woesearchaeales	B29-G15	B29-G15	B29-G15	sp003650585
UBA119	H	Woesearchaeales	UBA119	UBA119	UBA119	sp002505945
B72_G16	H	Woesearchaeales	B72-G16	B72-G16	B72-G16	sp003650545
CSSed11_301m	H	Woesearchaeales	CG1-02-33-12	PWVO01	PWVO01	sp003561825
CG_4_10_14_0_2_um_filter_33_10	H	Woesearchaeales	CG1-02-33-12	CG1-02-33-12	CG1-02-33-12	sp002762865
CG10_big_fil_rev_8_21_14_0_10_33_12	H	Woesearchaeales	CG1-02-33-12	CG1-02-33-12	CG1-02-33-12	sp002762865
CG06_land_8_20_14_3_00_33_13	H	Woesearchaeales	CG1-02-33-12	CG1-02-33-12	CG1-02-33-12	sp002762865
CG1_02_33_12	H	Woesearchaeales	CG1-02-33-12	CG1-02-33-12	CG1-02-33-12	sp002762865
FT1_394	I	Woesearchaeales	ARS49			
MP5_1_023	I	Woesearchaeales	ARS49			
SURF_65	I	Woesearchaeales	ARS49	SURF-65	SURF-65	sp003599055
J152	I	Woesearchaeales	ARS49	J152	J152	sp003694525
CG10_big_fil_rev_8_21_14_0_10_37_12	I	Woesearchaeales	ARS49	1-14-0-10-37-12	1-14-0-10-37-12	sp002762795
B103_G9	I	Woesearchaeales	ARS49			
ex4484_78	I	Woesearchaeales	ARS49			
MP5_1_69	I	Woesearchaeales	ARS49			
B100_G9	I	Woesearchaeales	ARS49			
Yap150.bin6.108	I	Woesearchaeales				
Yap100.bin1.6	I	Woesearchaeales				
FT1_425	I	Woesearchaeales				
Yap100.bin3.69	I	Woesearchaeales				
Yap150.bin7.227	I	Woesearchaeales				
Yap200.bin2.134	I	Woesearchaeales				
Yap100.bin4.218	I	Woesearchaeales				
Yap150.bin4.9	I	Woesearchaeales				
CG10_big_fil_rev_8_21_14_0_10_30_7	I	Woesearchaeales				
ARS49	I	Woesearchaeales	ARS49	GCA-2687275	GCA-2687275	sp002687275
YT1_142	I	Woesearchaeales	J091			
MP5_1_678	I	Woesearchaeales	B72-G16			
FT1_799	J	Woesearchaeales	B72-G16			
MP5_4_70	J	Woesearchaeales	B72-G16			
FT2_348	J	Woesearchaeales	B72-G16			
YT2_166	J	Woesearchaeales	B72-G16			
UBA10200	J	Woesearchaeales	UBA10200	UBA10200	UBA10200	sp10200u
ARS102	J	Woesearchaeales	GW2011-AR4	GCA-2688925	GCA-2688925	sp002688925
FT1_083	J	Woesearchaeales	GW2011-AR4	GCA-2686855		
ARS1041	J	Woesearchaeales	GW2011-AR4	GCA-2688925		
NP1295	J	Woesearchaeales	GW2011-AR4	GCA-2688925		
ARS74	J	Woesearchaeales	GW2011-AR4	GCA-2686855	GCA-2686855	sp002686855
ARS106	J	Woesearchaeales	GW2011-AR4	GCA-2688775	GCA-2688775	sp002688775
UBA12027	J	Woesearchaeales	GW2011-AR4	GW2011-AR11	GW2011-AR11	sp12027u
GW2011_AR11	J	Woesearchaeales	GW2011-AR4	GW2011-AR11	GW2011-AR11	sp12027u
UBA9638	J	Woesearchaeales	UBA11716	UBA11716	UBA11716	sp9638u
YT1_182	J	Woesearchaeales	UBA11716			
Yap2000.bin4.8	J	Woesearchaeales	UBA11716			
MP5_5_87	J	Woesearchaeales	UBA11716			
YT1_767	J	Woesearchaeales	UBA11716			
B3_Woes	J	Woesearchaeales	GCA-2687795	B3-WOES	B3-WOES	sp005222965
ARS1419	J	Woesearchaeales	GCA-2687795	GCA-2687795	GCA-2687795	sp002687795
Yap30.bin3.42	J	Woesearchaeales	B54-G15	B54-G15	B54-G15	sp003648985
B54_G15	J	Woesearchaeales	B54-G15	B54-G15	B54-G15	sp003648985
B66_G1	J	Woesearchaeales	B54-G15	B54-G15		
B43_G17	J	Woesearchaeales	B54-G15	B54-G15		
YT1_286	J	Woesearchaeales				
YT1_562	J	Woesearchaeales				
YT2_768	J	Woesearchaeales				
YT3_932	J	Woesearchaeales				
Yap2000.bin9.104	J	Woesearchaeales				
J072	Unassigned	Woesearchaeales	J072	J072	J072	sp003695265
Yap5000.bin1.9	Unassigned	Woesearchaeales	B72-G16			

REVIEWER COMMENTS

Reviewer #1 (Remarks to the Author):

Many thanks to the authors for the extensive edits, detailed comments and especially for providing all the material either in the supplements or in a repository, this is very much appreciated. The edits have overall approved the manuscript and I only have a few remaining comments.

1. Line 688-692: Was the ALE analysis done with the individual single gene tree files or with the bootstrap replicates? ALE needs either the bootstrap files (from maximum likelihood analyses and usually written by Iqtree if the wbt option is used) or mcmc samples (from Bayesian samples) to properly be able to estimate the probabilities of the gene trees ((as described in the github tutorial; <https://github.com/ssolo/ALE>)) and it is not completely clear to me what files were used.

Minor comments:

2. As a general comment:

Some of the newly added sentences might need some checks for sentence structure. For example in the sentence `It is notable that a higher peak of neutral proteins (with pIs ranging from 6 to 8) was observed in non-saline Woese archaeota than those in the saline group, while this peak is evident in outer-membrane proteomes` (Line 152) the side sentences are a bit intermingled making the text difficult to read.

3. Line 275: Instead of "to sum" maybe "to summarize" would be better

4. Line 294: Can the authors be more specific about the permease that was found? Based on the table I would assume it is a branched amino acid permease (livH)? Was also the substrate (livK) and ATP-binding domain (livG) of this permease found?

5. Line 665: Is this custom script available somewhere?

6. Line 1082: Phylogenetic tree of of Woese archaeota should be Phylogenetic tree of Woese archaeota

7. Line 1182: It should be Phylogenetic

Reviewer #2 (Remarks to the Author):

The revised version of this manuscript shows some improvement. In particular, some aspects of genome prediction are better detailed and data from the 16S rRNA gene environmental survey has been linked to MAG-derived information, which is useful. Nonetheless, I still have some concerns, and several queries that the authors have ignored in their response, as follows.

Salinity and ecological interpretation. All this discussion is poor and not well informed. First, the authors must define what they understand by "saline". If saline refers only to marine or brackish waters, we are far from the situation of strong adaptation to halophily – where pI values make a real difference and salt-in strategies might make sense. It is not expected that marine archaea have salt-in strategies, because marine waters (~3.5% salt) are not excessively challenging from this point of view (wide diversity of bacteria, archaea and eukaryotes). The freshwater-marine barrier has been known for a long time, but there are many organisms across the tree of life that can transition easily between the two. This is distinct from the adaptations to more saline environments. Consequently, if the authors want to have this discussion, they should concentrate on the marine-freshwater transition and forget about the 'salt-in'/'salt-out' strategy, that only applies to far higher salt concentrations than those apparently included in this study. If I am

wrong, please, do provide salinity gradients according to different levels of halophily as per the wide existing literature in the field. More specifically:

- Saline. This term applies to biotopes, not to organisms. 'Saline' Woese archaeota does not make sense.

- Saline – salinity. Define 'saline' in the context of this manuscript. Provide values along a gradient, e.g. freshwater, brackish, marine, 6-15% salt (halotolerant), >15% (halophiles), >20-25% hyperhalophiles. Salt-in strategies are only expected for halophilic archaea (and some bacteria, such as *Salinibacter*), i.e. for >15% salt. If the discussion refers to lower salinity levels, that discussion, which is not well-informed and fuzzy, should be eliminated.

- Lines 128-129. The authors conclude that salinity is the most important factor shaping Woese archaeota distribution and community. That might potentially be the case, as is for many other organisms that are adapted to freshwater or marine systems. However, they do not seem to test any other environmental parameter, such that this affirmation is only based on some kind of intuition rather on strict ecological testing. What about oxygen levels – these organisms seem to be present mostly in sediments and soils, where DO should be low. Or about pH or nutrients. In the absence of any serious ecological testing, the discussion based on halophilic adaptation is extremely superficial and not necessarily supported by ecological testing.

Other points:

- Title "...and a tendency toward an independent lifestyle". This is not actually reflected by their data. The vast majority of Woese archaeota genomes are quite reduced and lack important genes for autonomy. Even in the case of group J, with larger genomes, the biosynthesis of lipids is not ensured. This implies that this group also derives lipids from some type of host to build their membranes. Therefore, they do not have an independent lifestyle. Having a richer metabolic repertoire does not imply independence. Therefore, it is impossible to say if the acquisition of those genes will eventually lead to an autonomous lifestyle or not. For the moment, this does not seem to be the case, such that this part of the title is speculative.

- Line 46. 'Archaea constitute a considerable fraction of the microbial biomass of Earth'. This does not mean anything. We do not know exactly how much biomass archaea represent but, by their relative abundance across metagenomes and metabarcoding datasets, their biomass seems very low as compared to bacteria and eukaryotes.

- The authors use the term Thaumarchaeota, but then the GTDB nomenclature. Please, be consistent, avoid mixing nomenclatures, or provide equivalences.

- The authors conclude that lineage J has increased genome size by acquiring genes from bacteria and that they have a wider metabolic repertoire. Can they exclude the possibility that those gains were ancestral to the Woese archaeota clade and lost secondarily in other woese archaeal groups?

Reviewer #3 (Remarks to the Author):

The author's revisions are satisfactory to me. The new text should be checked throughout for grammar, however. Examples: Line 149-152; Line 139: is 1.5% the median or average? Check sentence.

REVIEWER COMMENTS

Reviewer #1 (Remarks to the Author):

Many thanks to the authors for the extensive edits, detailed comments and especially for providing all the material either in the supplements or in a repository, this is very much appreciated. The edits have overall approved the manuscript and I only have a few remaining comments.

1. Line 688-692: Was the ALE analysis done with the individual single gene tree files or with the bootstrap replicates? ALE needs either the bootstrap files (from maximum likelihood analyses and usually written by Iqtree if the wbt option is used) or mcmc samples (from Bayesian samples) to properly be able to estimate the probabilities of the gene trees ((as described in the github tutorial; <https://github.com/ssolo/ALE>)) and it is not completely clear to me what files were used.

Response: Thank you. We updated this analysis with the ultrafast bootstrap trees and added more information to the method. See line 339-345, 629.

Minor comments:

2. As a general comment:

Some of the newly added sentences might need some checks for sentence structure. For example in the sentence ` It is notable that a higher peak of neutral proteins (with pIs ranging from 6 to 8) was observed in non-saline Woese archaeota than those in the saline group, while this peak is evident in outer-membrane proteomes ` (Line 152) the side sentences are a bit intermingled making the text difficult to read.

Response: After our careful consideration, this part was removed given to reviewer #2 concerns.

3. Line 275: Instead of “to sum” maybe “to summarize” would be better

Response: Replaced as suggested.

4. Line 294: Can the authors be more specific about the permease that was found? Based on the table I would assume it is a branched amino acid permease (livH)? Was also the substrate (livK) and ATP-binding domain (livG) of this permease found?

Response: Thank you! The permease identified is not livH and it belongs to the amino acid-polyamine-organocation superfamily. To be more specific, we added more description at Line 278-279.

5. Line 665: Is this custom script available somewhere?

Response: This script was available at <https://doi.org/10.6084/m9.figshare.14459535>.

6. Line 1082: Phylogenetic tree of Woesearchaeota should be Phylogenetic tree of Woesearchaeota

Response: Corrected.

7. Line 1182: It should be Phylogenetic

Response: Corrected.

Reviewer #2 (Remarks to the Author):

The revised version of this manuscript shows some improvement. In particular, some aspects of genome prediction are better detailed and data from the 16S rRNA gene environmental survey has been linked to MAG-derived information, which is useful. Nonetheless, I still have some concerns, and several queries that the authors have ignored in their response, as follows.

Salinity and ecological interpretation. All this discussion is poor and not well informed. First, the authors must define what they understand by “saline”. If saline refers only to marine or brackish waters, we are far from the situation of strong adaptation to halophily – where pI values make a real difference and salt-in strategies might make sense. It is not expected that marine archaea have salt-in strategies, because marine waters (~3.5% salt) are not excessively challenging from this point of view (wide diversity of bacteria, archaea and eukaryotes). The freshwater-marine barrier has been known for a long time, but there are many organisms across the tree of life that can transition easily between the two. This is distinct from the adaptations to more saline environments. Consequently, if the authors want to have this discussion, they should concentrate on the marine-freshwater transition and forget about the ‘salt-in’/‘salt-out’ strategy, that only applies to far higher salt concentrations than those apparently included in this study. If I am wrong, please, do provide salinity gradients according to different levels of halophily as per the wide existing literature in the field. More specifically:

- Saline. This term applies to biotopes, not to organisms. ‘Saline’ Woesearchaeota does not make sense.

- Saline – salinity. Define ‘saline’ in the context of this manuscript. Provide values along a gradient, e.g. freshwater, brackish, marine, 6-15% salt (halotolerant), >15% (halophiles), >20-25% hyperhalophiles. Salt-in strategies are only expected for halophilic archaea (and some bacteria, such as *Salinibacter*), i.e. for >15% salt. If the discussion refers to lower salinity levels, that discussion, which is not well-informed and fuzzy, should be eliminated.

Response: Thank you very much for your nice comments and suggestions. After our careful consideration, we decided to remove the results and discussion about the adaptation to high salt, so that we can avoid the misleading.

- Lines 128-129. The authors conclude that salinity is the most important factor shaping Woesearchaeota distribution and community. That might potentially be the case, as is for many other organisms that are adapted to freshwater or marine systems. However, they do not seem to test any other environmental parameter, such that this affirmation is only based on some kind of intuition rather on strict ecological testing. What about oxygen levels – these organisms seem to be present mostly in sediments and soils, where DO should be low. Or about pH or nutrients. In the absence of any serious ecological testing, the discussion based on halophilic adaptation is extremely superficial and not necessarily supported by ecological testing.

Response: Thank you. We removed the results and discussion about the adaptation to high salt and modified the statement at Line 126-127.

Other points:

- Title “...and a tendency toward an independent lifestyle”. This is not actually reflected by their data. The vast majority of Woesearchaeota genomes are quite reduced and lack important genes for autonomy. Even in the case of group J, with larger genomes, the biosynthesis of lipids is not ensured. This implies that this group also derives lipids from some type of host to build their membranes. Therefore, they do not have an independent lifestyle. Having a richer metabolic repertoire does not imply independence. Therefore, it is impossible to say if the acquisition of those genes will eventually lead to an autonomous lifestyle or not. For the moment, this does not seem to be the case, such that this part of the title is speculative.

Response: Thank you! We agreed with the reviewer and dropped this part of the title.

- Line 46. ‘Archaea constitute a considerable fraction of the microbial biomass of Earth’. This does not mean anything. We do not know exactly how much biomass archaea represent but, by their relative abundance across metagenomes and metabarcoding datasets, their biomass seems very low as compared to bacteria and eukaryotes.

Response: Removed.

- The authors use the term Thaumarchaeota, but then the GTDB nomenclature. Please, be consistent, avoid mixing nomenclatures, or provide equivalences.

Response: Thank you. We made changes to keep the nomenclature consistent. Please see Line 53, 66, 220.

- The authors conclude that lineage J has increased genome size by acquiring genes from bacteria and that they have a wider metabolic repertoire. Can they exclude the possibility that those gains were ancestral to the Woesearchaeota clade and lost secondarily in other woesearchaeal groups?

Response: Thank you! Our statement did not rule out this possibility.

Reviewer #3 (Remarks to the Author):

The author's revisions are satisfactory to me. The new text should be checked throughout for grammar, however. Examples: Line 149-152; Line 139: is 1.5% the median or average? Check sentence.

Response: Thank you! We ran a thorough check on the grammar and made relevant changes. See Line 55, 136, 260, 276, 365, 393, 415, 425-427.

REVIEWERS' COMMENTS

Reviewer #1 (Remarks to the Author):

Thanks to the authors for integrating all my comments, I have no further things to add.

Reviewer #2 (Remarks to the Author):

I appreciate that Huang and colleagues have now withdrawn the highly speculative discussion on salinity adaptation from their manuscript. There are still some places, e.g. line 495, where they make statements that cannot be concluded from their work, such as 'salinity shapes the Woesearchaeota community'. They do not show that. Furthermore, in their figure 1, although the relative diversity of Woesearchaeota seem high in salt marshes (has this measure been normalized by rarefaction across datasets, by the way?), it seems rather low in the neighbouring "saline environment" (again, what salinity does this represent? This is stated nowhere in their manuscript). Therefore, from their analysis it is not particularly clear that salinity "shapes" Woesearchaeota communities. Salinity must certainly influence the distribution of Woesearchaeota, as it does for the rest of organisms, but whether this parameter is more influencing than others and explains their observed distribution remains to be demonstrated.

Lines 41-43 and, Woesearchaeota appears twice in the same sentence. Rephrase.

REVIEWER COMMENTS

Reviewer #1 (Remarks to the Author):

Thanks to the authors for integrating all my comments, I have no further things to add.

Response: Thank you.

Reviewer #2 (Remarks to the Author):

I appreciate that Huang and colleagues have now withdrawn the highly speculative discussion on salinity adaptation from their manuscript. There are still some places, e.g. line 495, where they make statements that cannot be concluded from their work, such as ‘salinity shapes the Woesearchaeota community’. They do not show that. Furthermore, in their figure 1, although the relative diversity of Woesearchaeota seem high in salt marshes (has this measure been normalized by rarefaction across datasets, by the way?), it seems rather low in the neighbouring “saline environment” (again, what salinity does this represent? This is stated nowhere in their manuscript). Therefore, from their analysis it is not particularly clear that salinity “shapes” Woesearchaeota communities. Salinity must certainly influence the distribution of Woesearchaeota, as it does for the rest of organisms, but whether this parameter is more influencing than others and explains their observed distribution remains to be demonstrated.

Response: Thanks. The measure of relative abundance and diversity of Woesearchaeota have been rarefied to the depth of 30,000. We agreed with the reviewer that analysis with specific salinity values is needed for the statement. Unfortunately, the salinity of many datasets is missing, including datasets attributed with the biotope “saline environment” and we have to determine whether a dataset is saline by the `emp_2` of the EMP ontology which was curated¹. To ease the tone and avoid confusion, we have modified the statement in Line 123, 363 and 368.

Lines 41-43 and, Woesearchaeota appears twice in the same sentence. Rephrase.

Response: We have modified the statement.

Reference:

1. Thompson, L. R. *et al.* A communal catalogue reveals Earth’s multiscale microbial diversity. *Nature* **551**, 457–463 (2017).